# Selective targeting of the TLR2/MyD88/NF-κB pathway reduces α-synuclein spreading in vitro and in vivo

Debashis Dutta[1,3], Malabendu Jana[1,3], Moumita Majumder[1], Susanta Mondal[1], Avik Roy[1] & Kalipada Pahan [1,2✉]

Pathways to control the spreading of α-synuclein (α-syn) and associated neuropathology in Parkinson's disease (PD), multiple system atrophy (MSA) and dementia with Lewy bodies (DLB) are unclear. Here, we show that preformed α-syn fibrils (PFF) increase the association between TLR2 and MyD88, resulting in microglial activation. The TLR2-interaction domain of MyD88 (wtTIDM) peptide-mediated selective inhibition of TLR2 reduces PFF-induced microglial inflammation in vitro. In PFF-seeded A53T mice, the nasal administration of the wtTIDM peptide, NEMO-binding domain (wtNBD) peptide, or genetic deletion of TLR2 reduces glial inflammation, decreases α-syn spreading, and protects dopaminergic neurons by inhibiting NF-κB. In summary, α-syn spreading depends on the TLR2/MyD88/NF-κB pathway and it can be reduced by nasal delivery of wtTIDM and wtNBD peptides.

---

[1] Department of Neurological Sciences, Rush University Medical Center, Chicago, IL, USA. [2] Division of Research and Development, Jesse Brown Veterans Affairs Medical Center, Chicago, IL, USA. [3] These authors contributed equally: Debashis Dutta, Malabendu Jana. ✉email: Kalipada_Pahan@rush.edu

Parkinson's disease (PD) is the most prevalent movement disorder affecting almost 6.1 million people worldwide[1]. The disease is caused by a specific loss of tyrosine hydroxylase (TH) positive dopaminergic (DAergic) neurons present in substantia nigra pars compacta (SNpc). One of the pathologic hallmarks of PD is the presence of Lewy bodies (LBs) containing aggregated α-syn. Lowering the deposition of aggregated α-syn from the brain parenchyma is expected to reduce the development and progression of not only sporadic and familial PD, but also dementia with Lewy bodies (DLB) and multiple system atrophy (MSA)[2–4].

It has been well manifested that point mutations (A53T, A30P, E46K, etc.) in the N-terminal region of α-syn or over-expression of the WT protein by gene duplication and triplication initiate amyloidogenesis[5], resulting in the formation of oligomeric and protofibrillar form of α-syn. Usually, α-syn aggregates are released from the neurons by exocytosis and the extracellular α-syn can spread into glial cells or neighboring neurons by cell-to-cell transfer or by non-cell-autonomous manner[6–9], inducing chronic inflammation in affected brain regions[10,11]. However, mechanisms by which α-syn spreading could be controlled are poorly understood. Activation of TLR2 requires its association with downstream adapter protein MyD88[12,13]. Recently, we have delineated that peptides corresponding to the TLR2-interacting domain of MyD88 (TIDM) specifically inhibit the induction of TLR2 activation without inhibiting either basal TLR2 activity or the activation of other TLRs[14].

Here, we demonstrated that preformed α-syn fibril (PFF) stimulated the interaction of TLR2 with its downstream partner MyD88 and that wild-type (wt) TIDM peptide reduced such interaction and decreased the generation of proinflammatory molecules in PFF-stimulated microglia without altering microglial phagocytic activity. Accordingly, the wtTIDM peptide inhibited α-syn spreading in the brain, prevented activation of glial cells, and attenuated parkinsonian pathologies in PFF-seeded sporadic model of PD. Furthermore, genetic deletion of TLR2 also halted the spreading of α-syn in the PFF-seeded brain, indicating an indispensable role of TLR2 in α-syn spreading. From the molecular level, we found that microglia-derived proinflammatory molecules increased neuronal α-syn expression via NF-κB-dependent transcriptional events and that blocking NF-κB activation in the brain by nasal NEMO-binding domain (NBD) peptide also decreased α-syn spreading and protected DAergic neurons in PFF-seeded A53T mice. These results suggest that the TLR2/MyD88/NF-κB pathway plays an important role in α-syn spreading and that targeting this pathway may have therapeutic importance for different α-synucleinopathies such as PD, MSA, and DLB.

## Results

**Inhibition of PFF-induced TLR2 activation by wtTIDM peptide**. The α-syn monomers were subjected to fibril formation under in vitro conditions over 7 days at 37 °C and then sonicated to produce short-length fibrils of the protein. Electron microscopy was performed to visualize the fibrillar structure of the protein (Fig. 1a). The data showed the thread-like structure of α-syn signifying successful conversion of the monomers into fibrils. Further SDS-PAGE and immunoblotting were performed with pure α-syn fibrils, where the presence of multimeric forms (38 kD and 50 kD) of the protein was observed (Fig. 1b). Next, to monitor PFF-induced TLR2 activation in microglia, mouse BV-2 microglial cells were treated with PFF, and TLR2-MyD88 interaction was evaluated by immunoprecipitation-coupled western blot. The findings exhibited remarkable enhancement in TLR2 and MyD88 interaction induced by PFF. However, this interaction was significantly inhibited by wtTIDM, but not by mTIDM, peptide (Fig. 1c, d and Supplementary Fig. S1). Since increased TLR2-MyD88 interaction leads to the activation of NF-κB, we monitored the effect of TIDM peptides on PFF-induced activation of NF-κB in BV-2 cells. Activation of NF-κB was monitored by both DNA-binding and transcriptional activities. While the DNA-binding activity of NF-κB was evaluated by the formation of a distinct and specific complex in electrophoretic mobility shift assay (EMSA), the transcriptional activity of NF-κB was monitored by the expression of luciferase from a reporter construct, pNF-κB-Luc. As evidenced from EMSA (Fig. 1e) and luciferase activity (Fig. 1f), PFF-treated BV-2 cells exhibited marked activation of NF-κB, and this was inhibited by wtTIDM, but not mTIDM, peptide. Since NF-κB is a proinflammatory transcription factor, we also monitored the expression of proinflammatory molecules such as inducible nitric oxide synthase (iNOS) and interleukin-1β (IL-1β), which are driven by NF-κB activation. Consistent to the induction of NF-κB activation, PFF alone led to marked mRNA expression of iNOS (Fig. 1g) and IL-1β (Fig. 1h) in primary microglia. However, wtTIDM, but not mTIDM, peptide markedly inhibited PFF-induced expression of iNOS and IL-1β in microglia (Fig. 1g, h). Overall, the findings indicate PFF-induced activation of TLR2 and consequent generation of inflammatory events in BV-2 cells and primary microglia can be attenuated by wtTIDM peptide.

**Microglial phagocytosis remains unaffected by TIDM**. Microglia serves as scavenging cells in the CNS for the degradation of extracellular debris and protein aggregates via phagocytosis. However, many anti-inflammatory molecules are known to inhibit microglial phagocytosis. Therefore, here, we examined the effect of TLR2 inhibition on phagocytosis. Phagocytic activity was monitored by using FITC-tagged monomeric α-syn in primary microglia isolated from WT and TLR2$^{-/-}$ mice. It is clearly evident that TLR2$^{-/-}$ microglia is less efficient in internalizing extracellular α-syn as compared to the WT microglia (Fig. 1i, j). These results are consistent with a previous study[15] showing that TLR2 ablation can compromise microglial phagocytosis. It further suggests that the presence of functional TLR2 is essential for phagocytosis as it is one of the receptors of α-syn. Interestingly, either wtTIDM- or mTIDM-treated WT microglia phagocytose extracellular α-syn to a similar extent as the untreated cells (Fig. 1k, m), suggesting that wtTIDM peptide does not alter microglial phagocytosis. Next, we monitored phagocytosis in LPS-stimulated microglia, where LPS was added to the cells 30 min after TIDM administration. As expected, we observed an increase in α-syn phagocytosis by LPS-stimulated microglia (Fig. 1l, m). However, either wtTIDM- or mTIDM-pretreated LPS-stimulated microglia also internalized a comparable amount of extracellular α-syn similar to only LPS-stimulated cells. To ensure that TIDM-mediated TLR2 inhibition does not hamper the microglial phagocytosis, we also performed this test using FITC-tagged fluorescence latex beads in both un-stimulated (Supplementary Fig. S2a) and LPS-stimulated (Supplementary Fig. S2b) primary microglia. Again, we found similar phagocytosis efficiency of external beads by microglia irrespective of TIDM treatment (Supplementary Fig. S2c). Although the absence of TLR2 hampered microglial phagocytosis, specific targeting of the interaction between intracellular MyD88 and TLR2 by wtTIDM peptide did not interfere with the phagocytic activity of microglia. These results suggest that selective inhibition of TLR2 activation by wtTIDM peptide can be an effective strategy to prevent PFF-induced microglial inflammation without hampering microglial phagocytosis.

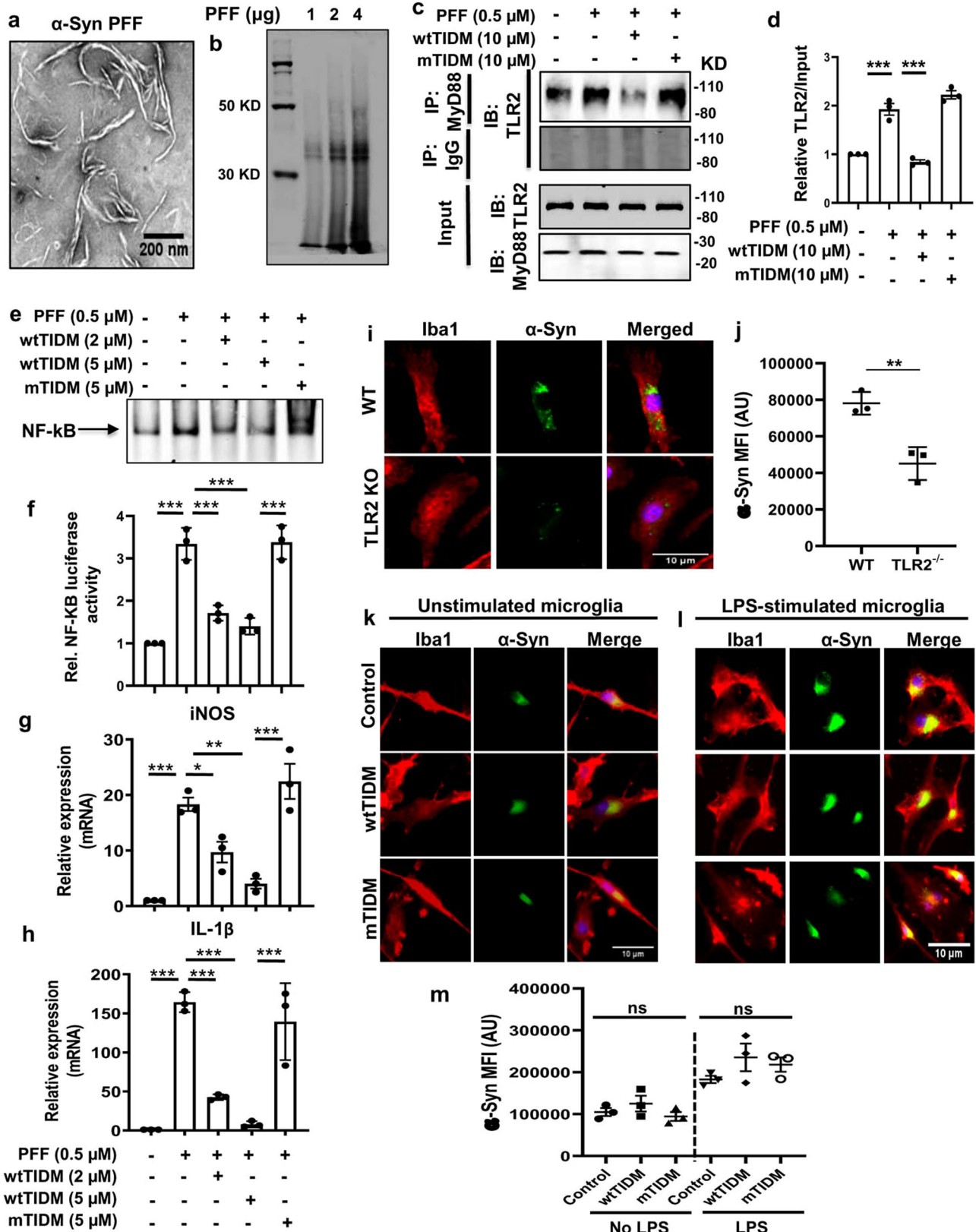

**Intranasal administration of wtTIDM peptide reduces gliosis in the SN of PFF-seeded mice**. Before testing wtTIDM peptide for controlling α-synucleinopathy, we monitored the level of TLR2 in the CNS of A53T mice. It was found that TH neurons (Supplementary Fig. S3a, c), Iba1-positive and P2RY12-positive microglia (Supplementary Fig. S3b, d–f) and GFAP-positive

astrocytes (Supplementary Fig. S3g, h) in the SN of A53T mice expressed a significantly higher level of TLR2 compared to the respective cell types in the non-transgenic (nTg) animals. Prior to conducting experiments in A53T animals, we evaluated the bioavailability of TIDM peptides in the brain of TIDM-treated B6C3 mice by using Alexa-680-labeled TIDM peptides. Infrared

**Fig. 1 Preformed α-syn fibril (PFF)-induced TLR2 activation in microglia.** Recombinant human α-syn was fibrillized in vitro to form preformed fibrils (PFF) and validated by electron microscopy to show fibrillar structure of the protein (**a**) and further by immunoblotting using anti-α-syn antibodies (**b**). BV-2 cells were preincubated with wtTIDM or mTIDM for 1 h followed by treatment with α-syn PFF (0.5 μM or 7 μg/ml), and the interaction of PFF-induced TLR2 and MyD88 was evaluated by immunoprecipitation (**c, d**, $p = 0.00012$ for control vs PFF and $p = 0.00039$ for PFF vs wtTIDM), NF-κB activation was measured in nuclear extracts by EMSA (**e**) and by luciferase assay in cells initially transfected with luciferase reporter gene constructs (**f**, $p = 0.00008$ for control vs PFF, $p = 0.00018$ for PFF vs wtTIDM 2 μM, $p = 0.000042$ for PFF vs wtTIDM 5 μM and $p = 0.000035$ for wtTIDM 5 μM vs mTIDM 5 μM). Wild type (WT) primary microglia, pretreated with wtTIDM or mTIDM, were exposed to PFF, RNA was isolated and the mRNA expression of induced nitric oxide synthase (iNOS, **g**, $p = 0.00031$ for control vs PFF, $p = 0.039$ for PFF vs wtTIDM 2 μM, $p = 0.0014$ for PFF vs wtTIDM 5 μM and $p = 0.00018$ for wtTIDM 5 μM vs mTIDM 5 μM) and interleukin-1β (IL-1β, **h**, $p = 0.00004$ for control vs PFF, $p = 0.0005$ for PFF vs wtTIDM 2 μM, $p = 0.00006$ for PFF vs wtTIDM 5 μM and $p = 0.00026$ for wtTIDM 5 μM vs mTIDM 5 μM) was quantified by real-time PCR. Microglia isolated from WT and TLR2$^{-/-}$ mice were treated with 0.5 μM of monomeric FITC-tagged α-syn for 2 h and the uptake of the protein was analyzed by immunofluorescence (**i, j**, $p = 0.0064$). WT microglia preincubated with 5 μM of either wtTIDM or mTIDM for 30 min were stimulated with LPS. After 1 h of LPS stimulation, monomeric FITC-tagged α-syn was added to the media, and phagocytosis was assessed by fluorescence analysis where FITC-tagged α-syn is shown within Iba1-positive microglia (**k, l**). MFI of intracellular α-syn was measured by ImageJ (**m**). Unpaired two-tailed *t*-test and one-way ANOVA followed by Tukey's multiple comparison tests were performed for statistical analyses. *$p < 0.05$, **$p < 0.01$, and ***$p < 0.001$ indicate significance compared to respective groups. Values are given as mean ± S.D. ($n = 3$ independent experiment).

scanning of mice demonstrated the presence of both wtTIDM and mTIDM peptides in the brain, although no signal was detected in the brain of only Alexa-680-treated animals, confirming the specificity of the experiment (Supplementary Fig. S4a). Furthermore, serial blocks of tissues were also scanned and showed remarkably higher availability of these peptides in the cerebellum, midbrain, and in the cortex (Supplementary Fig. S4b, c). Next, we monitored the effect of wtTIDM peptide in PFF-seeded A53T mice. However, unlike other studies where M20 mice[16,17] or M83$^{+/-}$ mice[18] were used for PFF seeding, we used M83$^{+/+}$ (A53T$^{+/+}$) mice for this experiment, because in the presence of a mutated form of the human α-syn gene in both the alleles, exogenous PFF-mediated pathology might develop much faster by homotropic species-specific interaction of α-syn[18]. The A53T animals start showing α-syn aggregation following 4 months of age, although no motor behavioral changes or dopaminergic cell death is found even in the later stage of the lifespan. Therefore, to induce a parkinsonian model, we injected PFF in the internal capsule (IC) region of striatum in both hemispheres of the mice (Supplementary Fig. S5a), and following 2 months of PFF seeding, animals received nasal delivery of 0.1 mg/kg/d of wtTIDM or mTIDM for the next 1 month. The total time span following PFF-injection was kept for 3 months based on earlier reports where researchers have shown the formation of α-syn pathology in SN following just one month of PFF induction[17]. The experimental animals were sacrificed at the age of 6 months and several biochemical tests were conducted from nigral and striatal tissues to find out the effect of TIDM treatment on PFF-induced pathology (Supplementary Fig. S5b). As wtTIDM peptide initially showed anti-inflammatory effects in PFF-exposed primary microglia, we examined the efficacy of this peptide in preventing glial inflammation in vivo in SN of PFF-seeded animals. It was found that nTg mice had a very low basal level of microglia (Supplementary Fig. S5c, d) and astroglia (Supplementary Fig. S7a, b) in the SN. Although some increase in the number of glial cells was observed in 6-month-old A53T mice receiving only PBS, the number of microglia and astroglia was found to be much greater in SN of PFF-seeded mouse brain (Supplementary Figs. S5c, d, and S7a, b). Increased number of glial cells also accompanied with a higher level of iNOS expression in both microglia and astroglia (Supplementary Figs. S5c, e and S7a, c) indicating aggravated gliosis and inflammation induced by PFF seeding. In addition, to further confirm PFF-induced microgliosis in SN, immunostaining was conducted for microglia-specific marker P2RY12 and the findings demonstrated an exaggerated number of P2RY12-positive cells in the SN of PFF-induced mice (Supplementary Fig. S6a, b). Interestingly, glial activation in terms of a number of glial cells as well as iNOS

expression was remarkably suppressed in SN of PFF-seeded mice treated with wtTIDM, but not mTIDM, peptide (Supplementary Figs. S5c, e, S6b, and S7a, c). Furthermore, the whole tissue protein level of important inflammatory glial markers such as Iba1, IL-1β, iNOS, and GFAP (Supplementary Fig. S5f–j) was assessed by immunoblotting and the level of these markers were also found to be significantly increased in SN of PFF-seeded mice as compared to PBS-injected A53T mice. However, intranasal treatment of PFF-seeded mice with wtTIDM, but not mTIDM, peptide led to suppression of these inflammatory molecules (Supplementary Fig. S5f–j).

**Intranasal administration of wtTIDM peptide reduces α-syn spreading from striatum to nigra and motor cortex in PFF-seeded mice.** Since glial activation occurs in parallel with the spreading of pathological α-syn[16] and wtTIDM inhibited PFF-induced glial activation in the SN, we examined the effect of TIDM on α-syn spreading in the brain. After PFF seeding in the IC, we observed a several-fold higher level of Triton X-100 insoluble form of α-syn in the SN as compared to the age-matched PBS-injected control group (Fig. 2b, d). However, intranasal wtTIDM peptide significantly decreased the detergent-insoluble form of α-syn in the SN, without exhibiting any visible change in the detergent soluble fraction (Fig. 2a, c). The finding was further substantiated by immunostaining of phosphoserine 129 form of α-syn (pSyn129) in the SN that demonstrated aggravated pSyn129 level in SN of PFF-seeded mice with both somatic and neuritic pathology. Again, the pSyn129 content was significantly reduced by treatment with wtTIDM, but not mTIDM (Fig. 2e, f). We also evaluated the level of pathological pSyn129 in the motor cortex and found the presence of thread-like fibrillar pSyn129 in cortical neurons of PFF-injected mice (Fig. 2g, h). However, parallel to SN, the level of pSyn129 drastically reduced in the motor cortex after wtTIDM administration (Fig. 2g, h). Total α-syn level in the motor cortex was evaluated by immunoblotting, which demonstrated the presence of higher molecular weight oligomeric forms in the detergent-insoluble protein fraction in PFF-injected mice (Fig. 2j, l) with no change in Triton X-100 soluble α-syn (Fig. 2i, k). Following wtTIDM administration, a significant reduction of monomeric and oligomeric α-syn was found only in detergent-insoluble fractions. On the other hand, mTIDM peptide failed to reduce insoluble α-syn in PFF-seeded brains (Fig. 2j, l). We also monitored pathological pSyn129 in the hippocampus (Supplementary Fig. S8a, c) and rostral striatum (Supplementary Fig. S8b, d) and found elevated level of pSyn129 in both the regions in PFF-seeded mice as evidenced by the presence of dark-stained pSyn129$^+$ neurons in the hippocampal CA1 region and dorsal striatum, which was

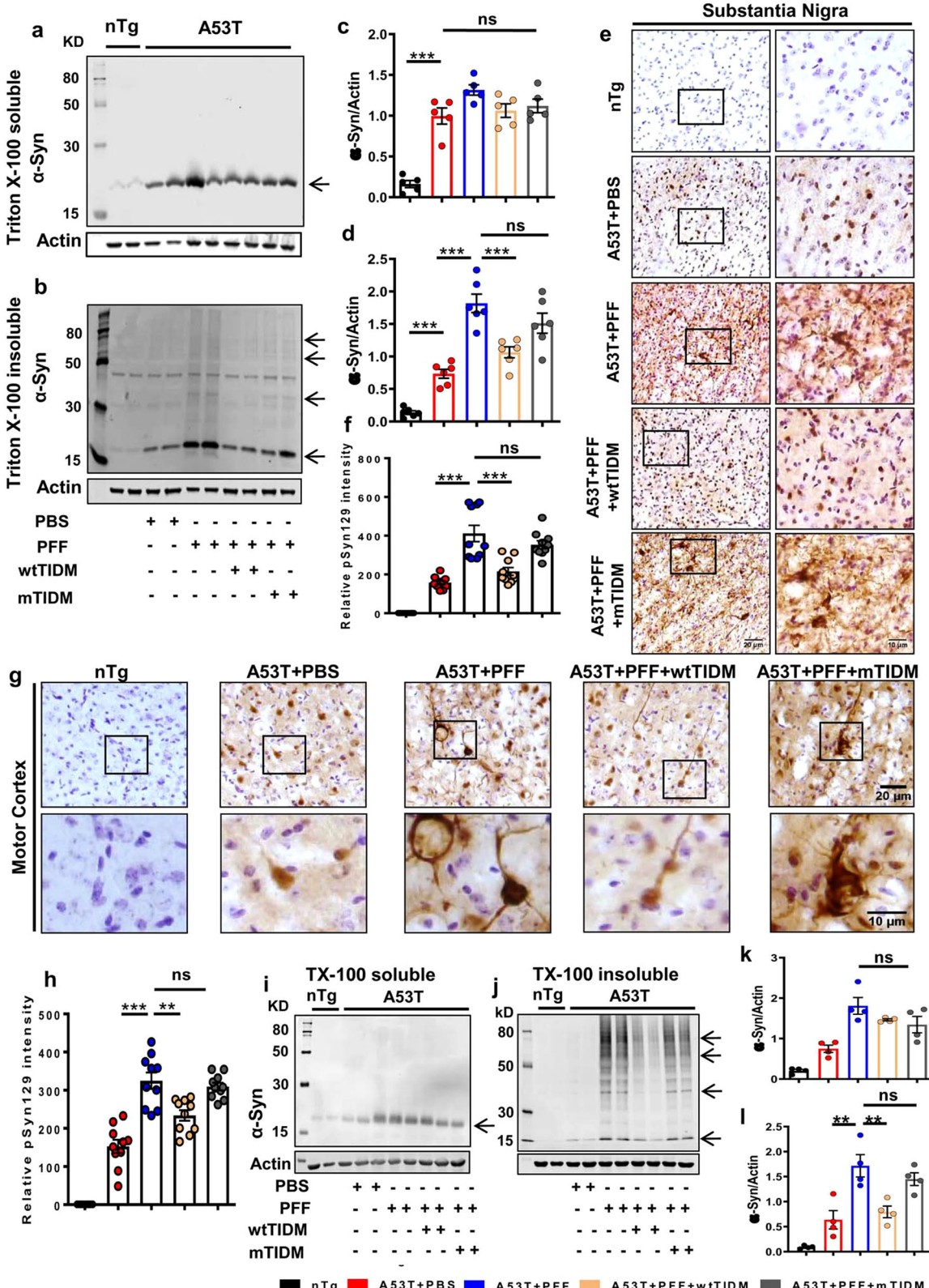

inhibited by treatment with wtTIDM, but not by mTIDM. Together, these results suggest that intranasal wtTIDM is capable of preventing α-syn spreading in the brain.

**Ablation of TLR2 reduces α-syn spreading in PFF-seeded mice.** To further confirm the involvement of TLR2 in α-syn spreading

in the brain, A53T mice lacking TLR2 (A53T$^{\Delta TLR2}$) were prepared by crossbreeding A53T$^{+/+}$ and TLR2$^{-/-}$ animals (Fig. 3a). Two months old A53T and A53T$^{\Delta TLR2}$ animals were seeded with PFF and following 3 months of brain surgery, experiments were performed. No significant change in soluble α-syn was found between A53T + PFF and A53T$^{\Delta TLR2}$ + PFF groups (Fig. 3b, d). The insoluble α-syn level was significantly higher in SN of PFF-

**Fig. 2 The wtTIDM peptide inhibits α-syn spreading from striatum to SN and motor cortex.** Spreading of α-syn in the brain of PFF-seeded A53T mice was monitored in SN region by immunoblotting of the protein isolated in Triton X-100 soluble (**a**) and insoluble (**b**) fractions. Arrows indicate the bands of different molecular weight of α-syn in insoluble fractions. Band density of monomeric α-syn in soluble (**c**, $n = 5$ animals, $p = 0.0002$ for nTg vs A53T + PBS) and insoluble (**d**, $n = 6$ animals, $p = 0.0002$ for A53T + PBS vs A53T + PFF and $p = 0.000336$ for A53T + PFF vs A53T + PFF + wtTIDM) fractions were normalized with the corresponding actin band. Level of phosphoserine 129 α-syn (pSyn129) in SN was evaluated by immunostaining followed by relative optical density measurement (**e**, **f**, $n = 5$ animals, $p = 0.00001$ for A53T + PBS vs A53T + PFF and $p = 0.00003$ for A53T + PFF vs A53T + PFF + wtTIDM). Propagation of α-syn in the motor cortex was monitored primarily by pSyn129 immunostaining (**g**, **h**, $n = 5$ animals, $p = 0.00001$ for A53T + PBS vs A53T + PFF and $p = 0.005$ for A53T + PFF vs A53T + PFF + wtTIDM) and immunoblotting of Triton X-100 soluble (**i**, **k**) and insoluble forms of the protein (**j**, **l**, $n = 4$ animals, $n = 4$ animals, $p = 0.0011$ for A53T + PBS vs A53T + PFF and $p = 0.0047$ for A53T + PFF vs A53T + PFF + wtTIDM). Two sections from each brain were taken for immunostaining, where Nissl was used for counterstaining. The value obtained from each section is shown in the graph. Images were deconvoluted by Fiji and DAB immunostaining specific to pSyn129 was measured and individual values obtained from each section are shown in the diagram. One-way ANOVA followed by Tukey's multiple comparison tests was conducted for statistical analyses. *$p < 0.05$, **$p < 0.01$, ***$p < 0.001$ indicate significance compared to respective groups. Values are given as mean ± SEM.

seeded A53T animals than in the PBS-injected group (Fig. 3c, e). However, detergent-insoluble α-syn content was significantly less in SN of A53T$^{\Delta TLR2}$ + PFF compared to A53T + PFF mice (Fig. 3c, e). The pronounced spreading of α-syn was validated further in SN of PFF-seeded A53T mice by pSyn129 staining. PFF-seeded A53T$^{\Delta TLR2}$ mice exhibited limited spreading in the SN region, and the accumulation of pSyn129 was significantly less than the PFF-seeded A53T mice (Fig. 3f, g). Simultaneously, A53T$^{\Delta TLR2}$ + PFF mice also showed remarkably less pSyn129 accumulation in the motor cortex than A53T + PFF mice (Fig. 3h, i). These findings clearly indicate that the spreading of pathological α-syn in different brain regions is significantly reduced in the absence of functional TLR2.

**Attenuation of α-syn spreading-induced DAergic neuronal death and PD pathologies by intranasal administration of wtTIDM.** PFF microinjection into IC was earlier shown to cause DAergic neuronal death in SN and it happens with gratuitous induction of gliosis[16,18]. Our findings have so far revealed that wtTIDM could effectively lower gliosis and reduce the spreading of pathological α-syn from striatum to nigra in PFF-seeded mice. Therefore, we monitored the vulnerability of DAergic neurons. However, prior to examining TH neuronal content in SN, we evaluated caspase 3 activation in nigral cells by immunostaining for cleaved caspase 3 to find out the induction of cell death triggered by α-syn in SN. We observed an exaggerated level of cleaved caspase 3 (Supplementary Fig. S9a, b, d), which correlated well with the enhanced aggregated level of α-syn in SN of PFF-induced animals (Supplementary Fig. S9a–c). In contrast, TH expression in surviving DAergic neurons of SN was significantly reduced in these animals (Supplementary Fig. S9b, e). This finding is of utmost importance as aggregated α-syn level parallels with enhanced caspase 3 and at the same time with decreased TH. In accordance with this finding, we found considerable loss of TH neurons in the nigra of PFF-seeded A53T mice as compared to PBS-microinjected mice. However, intranasal administration of wtTIDM, but not mTIDM, peptide significantly attenuated α-syn aggregation, caspase 3 activation, and the death of TH neurons (Supplementary Fig. S9). Parallel to this finding, the whole tissue TH expression was also found to be much higher in wtTIDM-treated PFF-seeded mice than either untreated or mTIDM-treated PFF-seeded mice (Fig. 4c, d). Next, we monitored TH fibers in the striatum that also revealed remarkable protection of TH fibers in wtTIDM-treated mice (Fig. 4e, f). This observation was further confirmed by western blot analysis (Fig. 4g, h). Since the restoration of TH cell body in SN and terminals in striatum should result in a greater level of DA and its metabolites 3,4-dihydroxyphenylacetic acid (DOPAC) and homovanillic acid (HVA), HPLC analyses for neurotransmitters were conducted from striatal tissues. As expected, PFF seeding caused around

40–50% loss of DA content in the striatum, which was significantly prevented by wtTIDM treatment (Fig. 4i–k). Finally, motor coordination of these animals was performed to check the functional outcome of wtTIDM-mediated restoration of nigral TH neurons and striatal DA level. Control 6 months old PBS-injected A53T animals did not show any motor abnormalities including rearing and in rotarod test compared to nTg group (Fig. 4l–p). However, compromised forepaw use and feet movement were clearly seen in PFF-seeded A53T mice, as evidenced by less number of rearing and poor performances in rotarod (Fig. 4l, m) and it is supported by earlier reports showing reduced forelimb and forelimb plus hindlimb activity of PFF-seeded animals[17,19]. Interestingly, wtTIDM treatment significantly corrected the behavioral abnormalities of PFF-seeded mice, whereas mTIDM-treated mice did not show any improvement in motor behavior. Overall, the findings indicate prevention of PFF-induced nigro-striatal pathology and improvement in motor behavior by intranasal wtTIDM treatment.

**The wtTIDM peptide prevents α-syn-induced pathology in aged A53T model of α-synucleinopathy.** After establishing the protective effect of wtTIDM against PFF-induced pathology, we extended the research in a more progressive model of α-synucleinopathy, A53T, which constitutively expresses the mutant form of human α-syn. At 9 months of age, A53T animals exhibit considerable glial activation and deposition of α-syn aggregates in the form of LBs in the affected regions of the brain including SN and hippocampus. These mice exhibit hyperactive behavior from 3 to 6 months of age, but the pathology becomes prominent with deficits in movement behavior at 9 months of age without demonstrating significant loss of nigral TH neurons[20]. Therefore, 8-months old A53T animals were given nasal delivery of wtTIDM and mTIDM (0.1 mg/kg daily) for the next one month followed by monitoring biochemical and behavioral parameters. Similar to the findings obtained from PFF-seeded mice, downregulation in the level of Triton X-100 insoluble pathological form of α-syn in SN was found in wtTIDM-administered animals, but not in an mTIDM-treated group (Supplementary Fig. S10c, d), although the soluble form of the protein was not altered by TIDM administration (Supplementary Fig. S10a, b). Moreover, the overall content and size of α-syn aggregates in nigral DAergic neurons were found to be significantly depleted by wtTIDM treatment in A53T animals (Supplementary Fig. S10e–h). Parallel to α-syn depletion, microglial inflammation was also inhibited as evidenced by lower expression of microglial iNOS (Supplementary Fig. S10i, j), decreased protein level of iNOS (Supplementary Fig. S10k, l) and IL-1β (Supplementary Fig. S10k, m) in the SN of a wtTIDM-treated group. In contrary, mTIDM-treated A53T animals did not exhibit any reduction in the level of proinflammatory molecules. Lastly, behavioral tests were conducted in

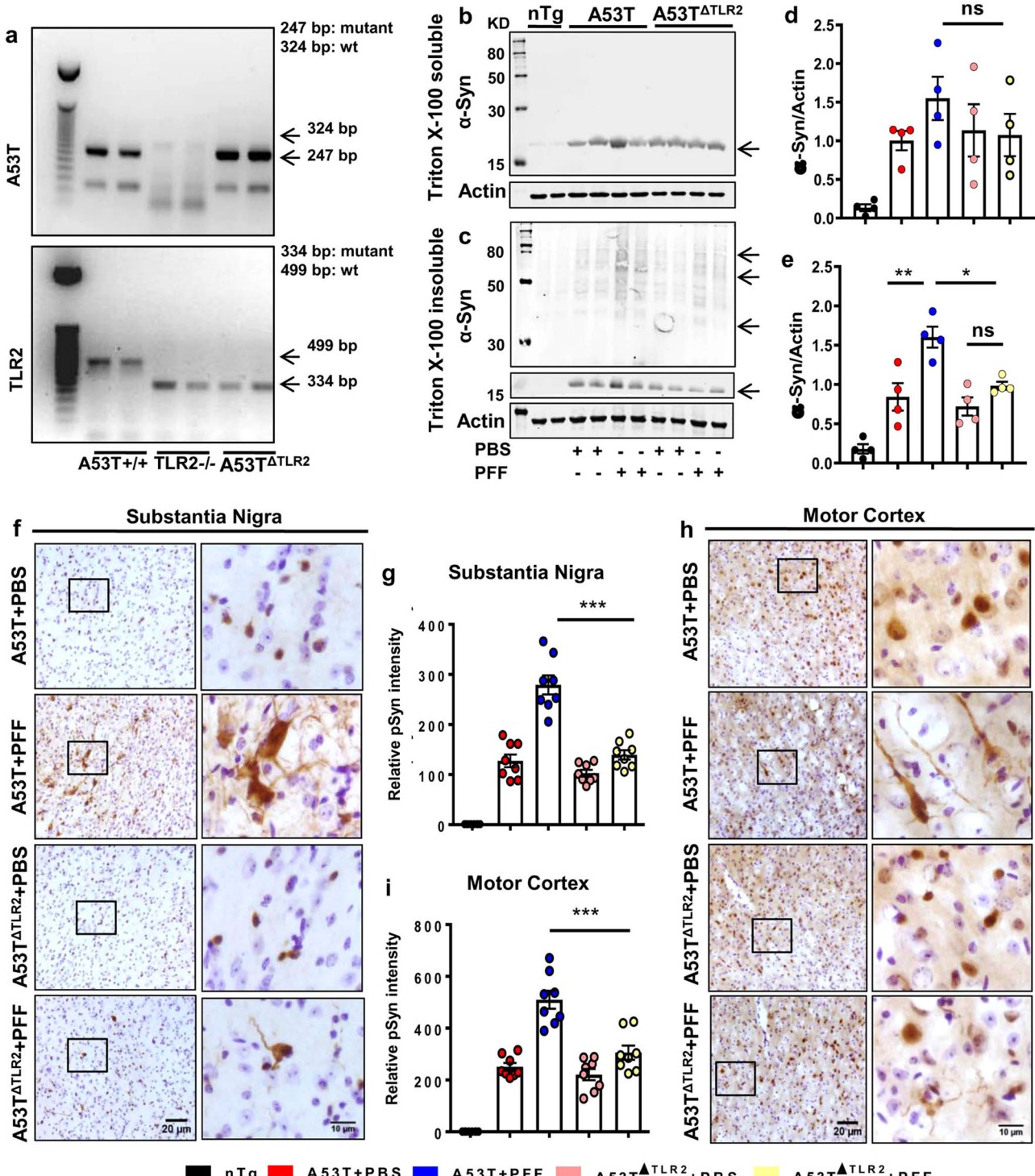

**Fig. 3 The α-Syn spreading is reduced in PFF-seeded A53T mice lacking TLR2.** A53T$^{+/+}$ mice were crossed with TLR2$^{-/-}$ mice to generate A53T$^{\Delta TLR2}$ double transgenic mice. These mice were validated by genetic screening where 324 and 247 bp bands correspond to nTg and A53T Tg mice respectively (marked with arrows). Similarly, 499 and 334 bp bands indicate nTg and TLR2$^{-/-}$ mice respectively (shown with arrows) (**a**). Spreading of α-syn in SN of PFF-seeded A53T and A53T$^{\Delta TLR2}$ mice was compared by assessing total α-syn level in Triton X-100 soluble (**b, d**) and insoluble (**c, e**, $p = 0.0027$ for A53T + PBS vs A53T + PFF and $p = 0.014$ for A53T + PFF vs A53T$^{\Delta TLR2+PFF}$) fractions by immunoblotting. Actin was used as the loading control. Level of pSyn129 in SN (**f, g**, $p = 0.00015$ for A53T + PFF vs A53T$^{\Delta TLR2+PFF}$) and motor cortex (**h, i**, $p = 0.0003$ for A53T + PFF vs A53T$^{\Delta TLR2+PFF}$) was monitored by immunostaining (**f, h**) followed by pSyn129 intensity analysis using Fiji. Two sections from each brain were used for staining and individual values from each section are shown. Two-way ANOVA was performed to determine statistical significance among different groups. *$p < 0.05$, **$p < 0.01$, and ***$p < 0.001$ indicate significance compared to respective groups. Values are given as mean ± SEM ($n = 4$ animals per group).

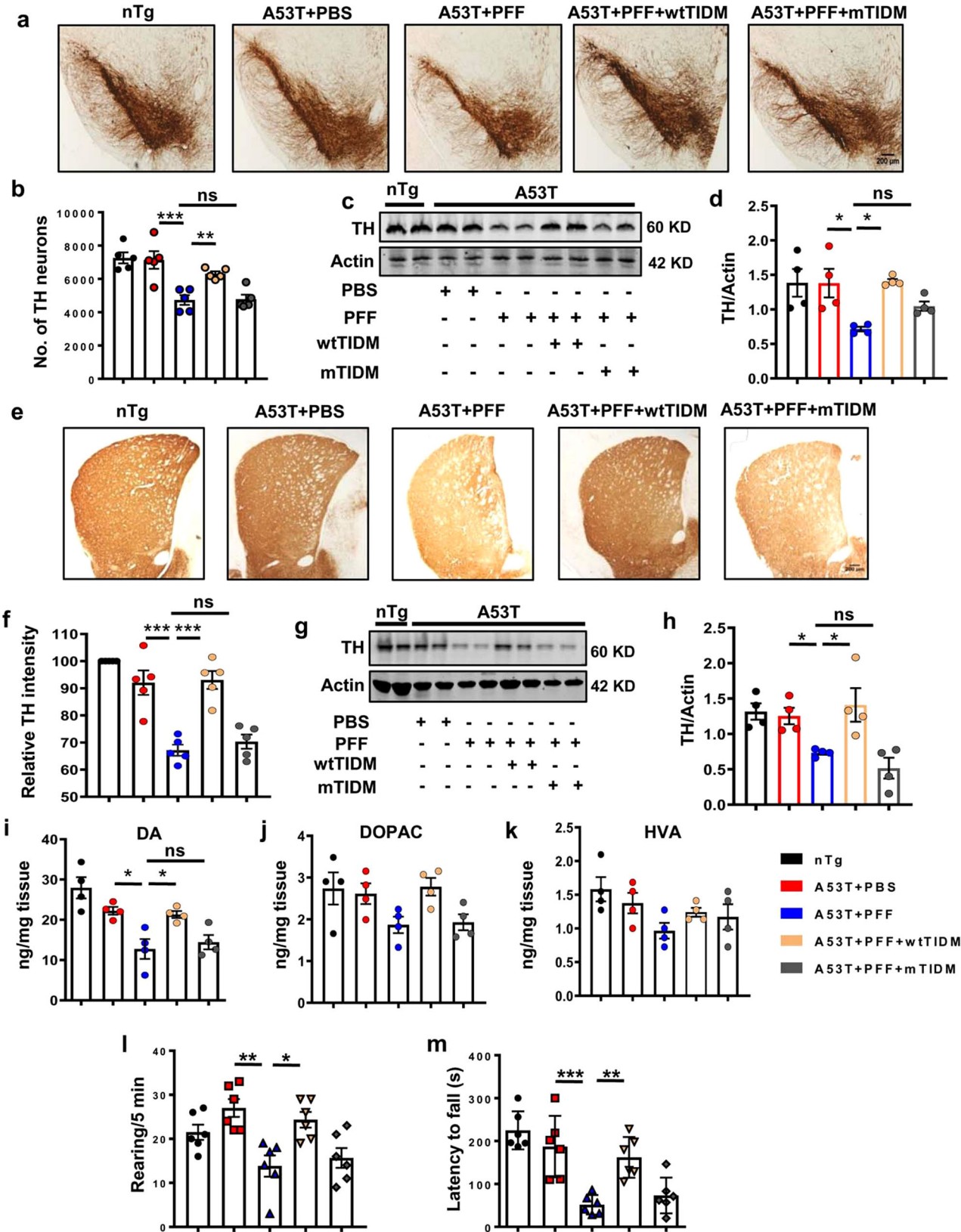

the aged A53T animals. As expected, A53T animals exhibited poor performance in the pole test as these mice took a longer time to turn downward (pole turn time) on the pole and to reach the base (climb downtime) of the pole (Supplementary Fig. S10n, o). However, consistent with the suppression of α-syn spreading,

wtTIDM treatment significantly improved the performance of A53T animals in the pole test.

**The wtTIDM peptide reduces α-synucleinopathy in A53T mice via TLR2**. To further confirm that wtTIDM exerts its anti-

**Fig. 4 Attenuated Parkinsonian pathology by wtTIDM in PFF-seeded A53T mice.** Nigral pathology in α-syn PFF-injected A53T control and wtTIDM/mTIDM administered mice was evaluated by immunohistochemical staining of TH (**a**) followed by stereological counting of viable TH + ve neurons in SN (**b**, n = 5 animals, p = 0.00052 for A53T + PBS vs A53T + PFF and p = 0.0025 for A53T + PFF vs A53T + PFF + wtTIDM). Images are shown at ×5 magnification, scale bar is 200 µm. TH protein level in SN was monitored by immunoblotting (**c**) and relative TH expression with respect to actin is shown (**d**, n = 4 animals, p = 0.0226 for A53T + PBS vs A53T + PFF and p = 0.0171 for A53T + PFF vs A53T + PFF + wtTIDM). Striatal pathology in experimental groups was checked by immunostaining of TH neuronal fibers in striatum (**e**). TH optical intensity was measured by Fiji and relative TH fiber intensity as a percentage to the nTg control brain is presented (**f**, n = 5 animals, p = 0.0005 for A53T + PBS vs A53T + PFF and p = 0.00034 for A53T + PFF vs A53T + PFF + wtTIDM). Images are shown at ×4 magnification. TH protein level in striatum is shown by immunoblotting (**g, h**, n = 4 animals, p = 0.0193 for A53T + PBS vs A53T + PFF and p = 0.0337 for A53T + PFF vs A53T + PFF + wtTIDM). Level of dopamine (DA, n = 4 animals, p = 0.0211 for A53T + PBS vs A53T + PFF and p = 0.0411 for A53T + PFF vs A53T + PFF + wtTIDM), 3, 4-dihydroxyphenylacetic acid (DOPAC, n = 4), and homovanillic acid (HVA, n = 4) was measured by HPLC-ECD method and the amount of neurotransmitters per mg of tissue was calculated (**i–k**). Behavioral performance of animals is demonstrated by a number of rearing (**l**, n = 6 animals, p = 0.0011 for A53T + PBS vs A53T + PFF and p = 0.0107 for A53T + PFF vs A53T + PFF + wtTIDM) and latency to fall down in rotarod test (**m**, n = 6 animals, p = 0.00045 for A53T + PBS vs A53T + PFF and p = 0.0044 for A53T + PFF vs A53T + PFF + wtTIDM). One-way ANOVA followed by Tukey's multiple comparison tests was conducted for statistical analyses. *p < 0.05, **p < 0.01, ***p < 0.001 indicate significance compared to respective groups. Values are given as mean ± SEM.

inflammatory and anti-proteinopathy effects via interaction with TLR2, 9-months-old A53T$^{\Delta TLR2}$ control and wtTIDM-treated A53T$^{\Delta TLR2}$ animals were compared with age-matched A53T mice with respect to the level of α-syn aggregates in neurons, the extent of gliosis, and lastly the motor behavior. In the absence of functional TLR2 protein, α-syn could not induce exaggerated microgliosis as evident from Iba1$^+$ microglia in the SN of A53T$^{\Delta TLR2}$ mice and age-matched A53T mice (Supplementary Fig. S11i, j). Reduced microgliosis was accompanied by significant downregulation of Triton X-100 insoluble form of α-syn in SN of A53T$^{\Delta TLR2}$ mice as compared to the A53T animals (Supplementary Fig. S11c, d). No significant downregulation of detergent soluble α-syn was found in A53T$^{\Delta TLR2}$ mice (Supplementary Fig. S11a, b). Similarly lower content of TH neuron-specific α-syn aggregates was found in SN of A53T$^{\Delta TLR2}$ mice (Supplementary Fig. S11e–h). Lack of glial inflammation in double transgenic animals was further confirmed by western blot analysis of iNOS (Supplementary Fig. S11k–l) and IL-1β (Supplementary Fig. S11k, m). Interestingly, in contrast to A53T mice, wtTIDM peptide remained unable to further decrease pathological features including α-syn aggregate level in midbrain tissues (Supplementary Fig. S11c, d), in TH neurons (Supplementary Fig. S11e–h) and proinflammatory molecules in A53T$^{\Delta TLR2}$ mice (Supplementary Fig. S11k, m). Finally, A53T$^{\Delta TLR2}$ mice performed significantly better than A53T mice in the pole test. However, wtTIDM peptide treatment did not improve movement activities of A53T$^{\Delta TLR2}$ mice (Supplementary Fig. S11n, o). These results suggest that in the absence of functional TLR2 protein, wtTIDM does not exhibit any neuroprotective effect in A53T mice.

**Proinflammatory cytokines upregulate the expression of α-syn in neuronal cells.** Next, we investigated molecular mechanisms by which TLR2 is coupled to α-synucleinopathy. We have convincingly showed that both microglial inflammation and spreading of pathological α-syn happen simultaneously in the brain of different models of α-synucleinopathy and that TLR2 is capable of controlling both pathological events. An increase in the level of α-syn is a prerequisite for α-synucleinopathy. However, it is not clear whether microglia-derived proinflammatory molecules directly modulate α-syn levels in neurons. Since activated microglia release different proinflammatory cytokines such as IL-1β and TNFα, mouse MN9D neuronal cells were treated with different concentrations of IL-1β and TNFα followed by monitoring the level of α-syn by western blot. The data demonstrated that both IL-1β (Fig. 5a, b) and TNFα (Supplementary Fig. S12a, b) increased α-syn protein level in a dose-dependent manner. Immuno-fluorescence analysis also showed that IL-1β (Fig. 5c) and TNFα (Supplementary Fig. S12c) were capable of increasing the level of α-syn protein in MN9D cells.

**Proinflammatory cytokines upregulate α-syn in mouse and human neuronal cells via NF-κB.** Next, we investigated mechanisms by which proinflammatory cytokines upregulate α-syn in neurons. Since proinflammatory cytokines induce the activation of NF-κB, we probed the α-syn gene promoter for NF-κB binding site by MatInspector search and found a consensus NF-κB binding site within −335 and −350 bp. Therefore, we examined the role of NF-κB activation in the expression of α-syn. The DNA-binding activity of NF-κB was evaluated by the formation of a distinct and specific complex in a gel shift DNA-binding assay. Treatment of MN9D cells with IL-1β resulted in the induction of DNA-binding activity of NF-κB (Fig. 5d). Similarly, IL-1β also induced the transcriptional activity of NF-κB as evident by luciferase activity from PBIIx-Luc construct with maximum activation seen at a concentration of 20 ng/ml (Fig. 5e). Next, to confirm the involvement of NF-κB in the expression of α-syn, we used wild type NF-κB essential modifier (NEMO) binding domain (wtNBD) peptide, known to specifically inhibit the induction of NF-κB activation[21,22]. Inhibition of α-syn mRNA expression by wtNBD, but not mNBD, peptide in neuronal cells treated with IL-1β (Fig. 5f) and TNFα (Supplementary Fig. S12d) suggests that proinflammatory cytokines stimulate α-syn expression in neurons via NF-κB.

Next, to test whether NF-κB transcriptionally regulates α-syn expression, the α-syn promoter (pα-syn-WT) containing the NF-κB binding site was cloned into the PGL3 enhancer vector (Fig. 5g). We also mutated the core NF-κB binding site and the mutated promoter construct (pα-syn-Mut) was cloned into the PGL3 vector (Fig. 5g). We found that IL-1β (Fig. 5h) and TNF-α (Supplementary Fig. S12e) markedly induced luciferase activity driven by wild type (pα-syn-WT), but not mutated (pα-syn-Mut), α-syn promoter. To further confirm the transcriptional regulation of α-syn promoter by NF-κB, we monitored the recruitment of NF-κB to the α-syn promoter by ChIP analysis. The classical NF-κB is a heterodimer of p65 and p50[21,22] and 1 h stimulation with IL-1β was sufficient to induce the recruitment of both p65 and p50 to the α-syn promoter in MN9D cells (Fig. 5i, j). Since histone acetyltransferases such as CREB-binding protein (CBP) and p300, play an important role in transcriptional activities, we investigated if CBP and p300 were also involved in IL-1β-induced transcription of the α-syn gene. As evident from PCR (Fig. 5i) and real-time PCR (Fig. 5j), IL-1β stimulation induced the recruitment of p300, but not CBP, to the α-syn gene promoter. Consistent with the recruitment of p65, p50, and p300, IL-1β was also able to recruit RNA polymerase (RNA Pol) to the α-syn gene promoter (Fig. 5i, j). These results are specific as we did not see any amplification product in any of the immuno-precipitates obtained with control IgG (Fig. 5i, j). It indicates that IL-1β is capable of inducing the

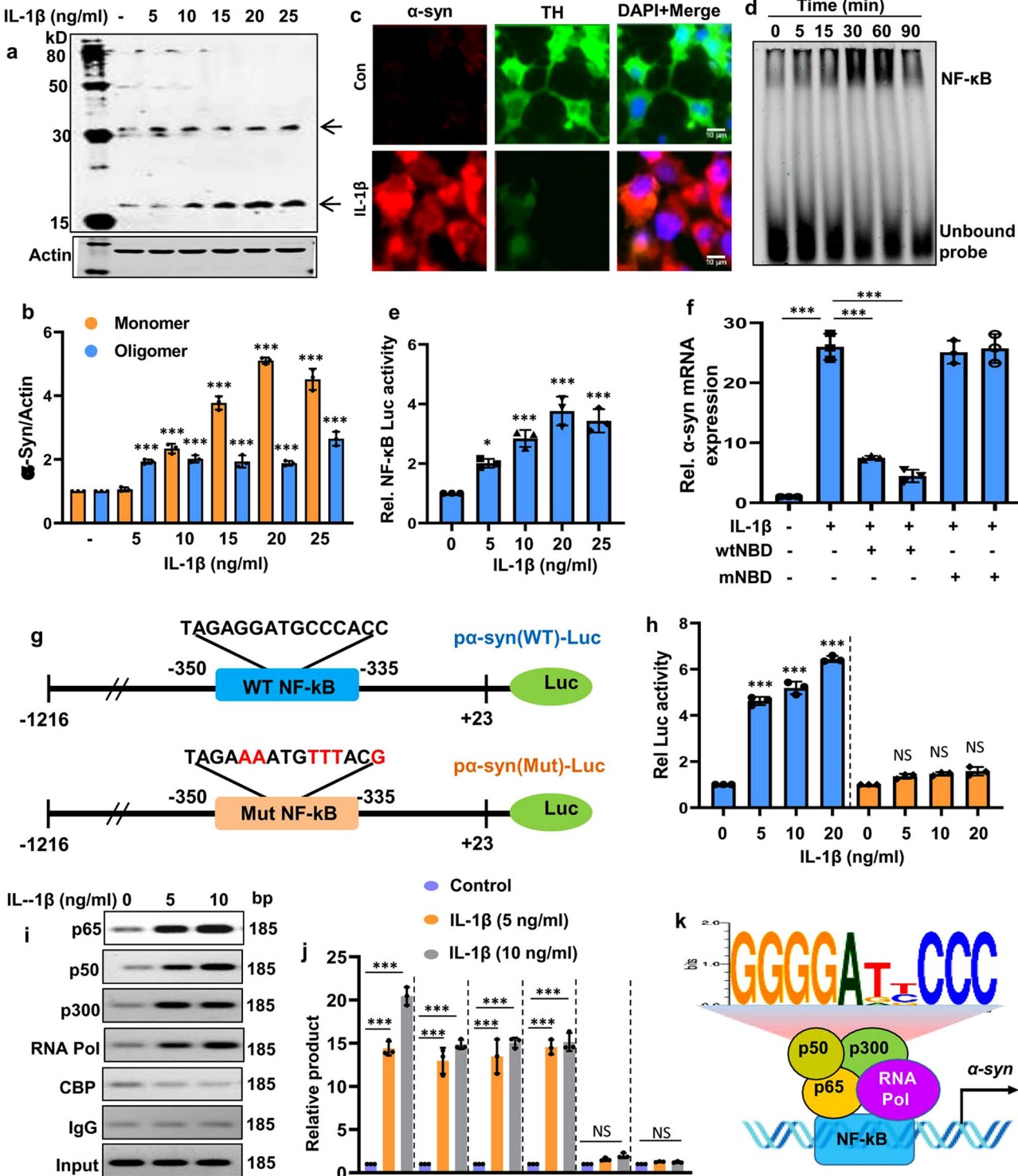

recruitment of p65, p50, p300, and RNA pol to mouse *α-syn* promoter in mouse MN9D neuronal cells (Fig. 5k).

Next, we examined whether NF-κB was also involved in IL-1β-induced expression of α-syn in human SH-SY5Y neuronal cells. Similar to mouse α-syn promoter, the human α-syn promoter also harbors a consensus NF-κB binding site (Supplementary Fig. S13a). Accordingly, IL-1β also induced the DNA-binding (Supplementary Fig. S13b) and transcriptional (Supplementary Fig. S13c) activities of NF-κB in SH-SY5Y cells. Inhibition of IL-1β-induced upregulation of monomeric and oligomeric forms of α-syn protein in SH-SY5Y cells by wtNBD, but not mNBD,

peptide suggests that similar to mouse neuronal cells, IL-1β also requires NF-κB to increase α-syn in human neuronal cells (Supplementary Fig. S13d–f). Finally, a significant amount of p65, p50, p300, and RNA Pol was found to be recruited to the human *α-syn* promoter in IL-1β-treated SH-SY5Y cells (Supplementary Fig. S13g, h), highlighting the importance of NF-κB in IL-1β-induced transcription of the human *α-syn* gene.

**Is activation of NF-κB sufficient for the induction of α-syn expression?** In addition to inducing the activation of NF-κB,

**Fig. 5 Activation of NF-κB increases α-syn expression in neurons.** Cells were stimulated with different concentrations of IL-1β under serum-free conditions. After 12 h of IL-1β treatment, the protein level of α-syn was examined by western blot (**a**, **b**, $p = 0.00001$ for α-syn monomer, control vs IL-1β 10–25 ng/ml doses and for α-syn oligomer, $p = 0.000023$ control vs IL-1β 5 ng/ml, $p = 0.00009$ control vs IL-1β 10 ng/ml, $p = 0.000021$ control vs IL-1β 15 ng/ml, $p = 0.00004$ control vs IL-1β 20 ng/ml, $p = 0.00001$ control vs IL-1β 25 ng/ml). After 12 h of IL-1β treatment, cells were also immunostained with antibodies against α-syn and TH (**c**). Mouse MN9D cells were incubated with IL-1β for different time periods followed by monitoring the activation of NF-κB by EMSA (**d**). MN9D cells were transfected with PBIIx-Luc for 24 h followed by treatment with different concentrations of IL-1β and subjected to luciferase assay (**e**, $p = 0.0183$ control vs IL-1β 5 ng/ml, $p = 0.00022$ control vs IL-1β 10 ng/ml, $p = 0.00006$ control vs IL-1β 20 ng/ml, $p = 0.00002$ control vs IL-1β 25 ng/ml). Cells preincubated with either wtNBD peptide or mNBD peptide for 30 min were stimulated by IL-1β for 4 h followed by the analysis of α-syn mRNAs by quantitative real-time PCR (**f**, $p = 0.00001$ control vs IL-1β, $p = 0.00001$ IL-1β vs IL-1β + wtNBD both doses). Map of wild type and mutated NF-κB site of α-syn-luciferase promoter constructs (**g**). MN9D cells were transfected with pα-syn(WT)-Luc and pα-syn(Mut)-Luc for 24 h followed by treatment with IL-1β and subjected to luciferase assay (**h**, $p = 0.00001$ for control vs IL-1β all doses). MN9D Cells were treated with IL-1β for 1 h in serum-free media. Then immunoprecipitated chromatin fragments were amplified by semi-quantitative (**I**), and quantitative PCR (**j**, $p = 0.00001$ or less for the indicated asterisks) for the indicated region spanning the proximal NF-κB of the *α-syn* promoter using primers mentioned under "Methods". ** and *** indicate $p < 0.01$ and $p < 0.001$ compared to the control. "NS" indicates not significant. One-way ANOVA followed by Tukey's multiple comparison test was used for statistical analyses. All results are presented as mean ± S.D. ($n = 3$ independent experiments). The schematic diagram depicts a detailed map of promoter analysis of *α-syn* gene (**k**). The map reveals a conserved NF-κB-responsive element in the promoter of *α-syn* gene at −350 to −335 upstream of the *α-syn* transcription start site.

proinflammatory molecules are known to transduce many other signaling pathways. Therefore, we examined whether activation of NF-κB alone is sufficient to induce the expression of α-syn in neuronal cells. The p65 (RelA) subunit of NF-κB, cloned in pcDNA3, was over-expressed in MN9D cells followed by monitoring the level of α-syn. It was found that the overexpression of p65 alone markedly increased the level of α-syn in MN9D neuronal cells (Supplementary Fig. S14a, b). These results were specific as the empty vector remained unable to increase α-syn in MN9D cells (Supplementary Fig. S14a, b). These results suggest that activation of NF-κB is sufficient to increase the expression of α-syn in neurons.

**Etiological reagents of different neurodegenerative disorders upregulate the activation of α-syn promoter via NF-κB.** Along with primary α-synucleinopathies (PD, DLB, and MSA), α-syn pathology is observed in various inflammatory conditions associated with viral encephalopathy, multiple sclerosis (MS), HIV-associated neurocognitive disorders (HAND), etc.[23,24]. Because IL-1β and TNFα upregulated the activation of α-syn promoter in neuronal cells, we were prompted to investigate whether different proinflammatory molecules were also capable of upregulating α-syn promoter. Therefore, MN9D neuronal cells transfected with wild type (*pα-syn-WT-Luc*) and mutated (*pα-syn-Mut-Luc*) α-syn promoter-driven reporter constructs were stimulated with dsRNA in the form of poly IC (one of the etiological reagents for viral encephalopathy), HIV-1 Tat (one of the etiological reagents for HAND), MPP$^+$ (Parkinsonian toxin), bacterial lipopolysaccharides or LPS (a prototype inflammatory stimulus), CpG oligonucleotide or ODN (bacterial infection-associated inflammation), and IFNγ (Th1 cytokine associated to MS). Similar to IL-1β and TNFα, MPP$^+$ (Supplementary Fig. S15a), LPS (Supplementary Fig. S15b), Tat (Supplementary Fig. S15c), ODN (Supplementary Fig. S15d), and poly IC (Supplementary Fig. S15e) markedly induced luciferase activity driven by wild type (*pα-syn-WT*), but not mutated (*pα-syn-Mut*), α-syn promoter. In contrast, IFNγ induced the activation of both wild type (*pα-syn-WT-Luc*) and mutated (*pα-syn-Mut-Luc*) α-syn promoters (Supplementary Fig. S13f). These results suggest that MPP$^+$, LPS, Tat, ODN, and poly IC, but not IFNγ, require NF-κB for the activation of *α-syn* promoter in neurons.

**Microglia-derived proinflammatory molecules upregulate α-syn expression in primary DAergic neurons.** After establishing the molecular mechanism behind inflammatory molecule-mediated α-syn transcriptional upregulation in MN9D and SH-SY5Y cells, we further validated the phenomenon in mouse primary DAergic neurons. In this case, primary microglia pretreated with wtTIDM or mTIDM peptide for 30 min were challenged with PFF. Following 24 h of PFF exposure, we measured microglia-secreted inflammatory molecules such as IL-1β and TNF-α from the spent medium by sandwich ELISA. The results showed a marked increase in the level of these cytokines in PFF-treated microglia compared to the untreated control. However, wtTIDM, but not mTIDM, peptide pre-treatment significantly reduced the level of microglia-secreted IL-1β and TNFα in the media (Supplementary Fig. S16a, b). Then to evaluate the effect of microglia-secreted inflammatory molecules on the expression of neuronal α-syn, we performed the co-culture analysis with primary microglia and midbrain DAergic neurons. As described above, microglia were treated with TIDM for 30 min followed by stimulation with PFF. After 12 h of PFF exposure, the insert containing the microglia was placed on top of the neuronal culture so that microglia-derived proinflammatory molecules can come in contact with the neurons (Fig. 6a). Firstly, following 3 h of microglia-neuron co-culture, the status of NF-κB activation was assessed in TH$^+$ DAergic neurons by immunostaining for Ser536 phospho-p65. The findings demonstrated significant upregulation of phospho-p65 following PFF exposure (Fig. 6b). Interestingly, neurons co-cultured with wtTIDM-treated microglia showed reduced expression of phospho-p65, but it was not found in the case of mTIDM treatment (Fig. 6b). Furthermore, we evaluated α-syn mRNA expression in primary neurons following 6 h of co-culturing with PFF-treated microglia. As expected, PFF-exposed neurons exhibited around 6-fold increase in α-syn expression compared to the control group. However, α-syn expression markedly decreased in PFF-insulted neurons upon treatment with wtTIDM, but not mTIDM (Fig. 6c). Lastly, α-syn protein expression was assessed by immunostaining in TH$^+$ neurons, where parallel to the α-syn mRNA expression, we found higher α-syn protein level following co-culturing with PFF-treated microglia. Interestingly, the α-syn protein level was markedly reduced in neurons cultured with wtTIDM-treated microglia (Fig. 6d, e). These data suggest that under PFF exposure, microglia-secreted proinflammatory molecules stimulate the expression of neuronal α-syn via activating NF-κB and that wtTIDM pre-treatment inhibits the generation of proinflammatory molecules by microglia resulting in reduced upregulation of neuronal α-syn expression. To further strengthen the role of NF-κB activation in mediating inflammation-induced upregulation of neuronal α-syn, we conducted another microglia-

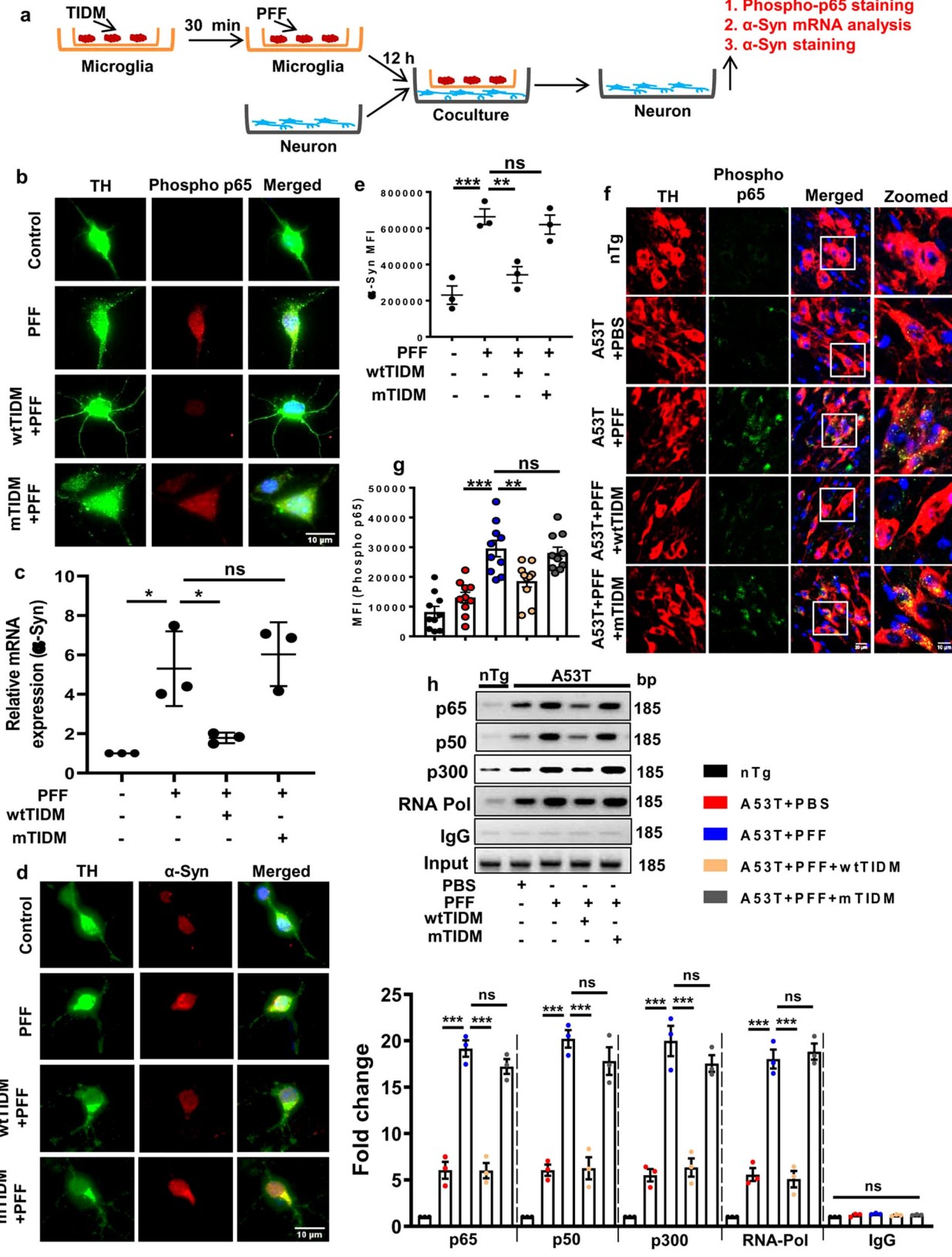

neuron co-culture experiment, where microglia was initially exposed to PFF for 12 h and then co-cultured with neurons, but 30 min prior to co-culturing, neurons were treated with either wtNBD or mNBD peptides (Supplementary Fig. S14c). Expression analysis for α-syn mRNA and protein in these neurons demonstrated a marked increase in PFF-exposed neurons. However, wtNBD-treated neurons could significantly attenuate the PFF-induced upregulation of α-syn expression that was not observed in mNBD-treated neurons (Supplementary Fig. S16d–f).

**Fig. 6 NF-κB-mediated upregulation of α-syn in primary DAergic neurons and NF-κB activation in vivo.** Primary microglia were treated with either wtTIDM or mTIDM (5 μM) and after 30 min challenged with PFF (7 μg/ml). Following 12 h of PFF exposure, the insert containing microglia was placed on to the culture dish containing primary DAergic neurons (**a**). After 3 h of co-culturing, immunostaining was performed for phospho Ser536 p65 in TH⁺ DAergic neurons (**b**). The mRNA expression of α-syn in neurons was monitored after 6 h of co-culturing by real-time PCR (**c**, n = 3 samples, p = 0.0128 for control vs PFF and p = 0.0363 for PFF vs wtTIDM + PFF). Protein expression of α-syn in neurons was assessed after 12 h of co-culturing by immunocytochemistry followed by MFI analysis using ImageJ and at least 10 cells from each group per experiment were measured for MFI analysis (**d**, **e**, n = 3 experiments, p = 0.00097 for control vs PFF and p = 0.0065 for PFF vs wtTIDM + PFF). Activation of NF-κB in nigral DAergic neurons of experimental mice was monitored by phospho Ser536 p65 staining in TH⁺ neurons followed by MFI analysis using ImageJ (**f**, **g**, n = 5 animals, p = 0.000014 for A53T + PBS vs A53T + PFF and p = 0.0054 for A53T + PFF vs A53T + PFF + wtTIDM). Two sections from each brain were taken for the immunofluorescence analysis and value obtained from each section is shown in the graph. Images were captured at ×60 magnification and further zoomed. Activation of α-syn promoter by NF-κB was assessed by ChIP analysis using antibodies for p65, p50, p300, and RNA pol II, whereas IgG was used as the negative control. Immunoprecipitated DNA fragments were amplified by real-time PCR using primers mentioned in the "Methods" section (**h**, n = 3 animals, p65, p = 0.000021 for A53T + PBS vs A53T + PFF and p = 0.000021 for A53T + PFF vs A53T + PFF + wtTIDM; p50, p = 0.000011 for A53T + PBS vs A53T + PFF and p = 0.000012 for A53T + PFF vs A53T + PFF + wtTIDM; p300, p = 0.000009 for A53T + PBS vs A53T + PFF and p = 0.000016 for A53T + PFF vs A53T + PFF + wtTIDM; RNA Pol, p = 0.000004 for A53T + PBS vs A53T + PFF and p = 0.000003 for A53T + PFF vs A53T + PFF + wtTIDM). One-way ANOVA followed by Tukey's multiple comparison tests was conducted for statistical analyses. *p < 0.05, **p < 0.01, ***p < 0.001 indicate significance compared to respective groups. Values are given as mean ± SD for cell culture analysis and mean ± SEM for in vivo analysis.

**The wtTIDM peptide prevents NF-κB activation in nigral DAergic neurons.** As wtTIDM was found to inhibit PFF-induced activation of NF-κB in primary DAergic neurons, it was necessary to evaluate whether the finding was also recapitulated in vivo. We assessed the level of Ser536 phospho-p65 expression in nigral neurons by immunofluorescence analysis. The data demonstrated a significant increase in phospho-p65 level in TH⁺ neurons of PFF-injected mice (Fig. 6f, g). However, activated p65 level in nigral neurons was markedly less in wtTIDM-treated PFF-induced animals that was not observed in the case of mTIDM-treated animals (Fig. 6f, g). To confirm the involvement of NF-κB in the expression of α-syn further, we examined the recruitment of NF-κB to *α-syn* gene promoter in vivo in the SN by in situ ChIP. We found marked enrollment of p65 and p50 to the *α-syn* gene promoter in the SN of PFF-treated mice as compared to PBS-injected mice (Fig. 6h, i). Parallel to this, the binding of p300 and RNA polymerase II to α-syn promoter was also found to be significantly elevated in the PFF-insulted brain with respect to the PBS-injected samples (Fig. 6h, i). It indicates transcriptional activation of *α-syn* gene via activation of classical NF-κB following PFF induction. Interestingly, the recruitment of p65, p50, p300, and RNA polymerase II was found to be greatly reduced in wtTIDM-treated, but not mTIDM-treated, PFF-insulted mice (Fig. 6h, i). Collectively, these results indicate that wtTIDM peptide inhibit NF-κB-mediated transcriptional activation of *α-syn* gene.

**Intranasal administration of NEMO-binding domain (NBD) peptide reduces α-syn spreading from striatum to nigra in PFF-seeded mice.** Findings presented above indicate that PFF-induced TLR2 activation causes the generation of microglial proinflammatory molecules to upregulate the expression of α-syn in neurons via NF-κB activation. This prompted us to investigate the role of NF-κB in α-syn spreading in the brain. The wtNBD peptide is a specific inhibitor of NF-κB activation[22] and we have demonstrated that after intranasal administration, wtNBD peptide enters into the brain[25]. Therefore, here, we examined the effect of intranasal wtNBD on α-syn spreading in the brain. Following 1 month of wtNBD administration, NF-κB activation in SN and spreading of α-syn in both SN and motor cortex were monitored. We found marked upregulation of microglial acetylated p65 level in the SN of PFF-seeded mice compared to the PBS-injected mice (Supplementary Fig. S17a, b). However, acetylated p65 level was significantly decreased in wtNBD-treated mice brain (Supplementary Fig. S17a, b). As described above, PFF

seeding resulted in exaggerated accumulation of pSyn129 in nigral neurons (Fig. 7a, b). However, intranasal wtNBD peptide drastically reduced the level of pSyn129 in these neurons, which is reflected by the relative optical density measurement of pSyn129 in SN (Fig. 7a, b). This observation was also verified by immunoblotting, where PFF-seeded mice exhibited the presence of a higher level of detergent-insoluble form of α-syn than PBS-injected mice (Fig. 7d, f). However, following wtNBD treatment α-syn contents in both soluble and insoluble fractions were significantly reduced (Fig. 7c–f). Similar to nigra, wtNBD peptide also reduced the spreading of α-syn in the motor cortex as evidenced by reduced accumulation of pSyn129 in cortical neurons of wtNBD-treated PFF-seeded mice as compared to saline-treated PFF-seeded mice (Fig. 7g, h).

**Intranasal NBD peptide protects DAergic neurons and improves locomotor activities in PFF-seeded mice.** Next, we monitored the effect of intranasal wtNBD on PFF-induced parkinsonian pathologies. Significant reduction in the number of TH neurons (Supplementary Fig. S18a) as well as nigral TH protein level (Supplementary Fig. S18b, c) was seen in PFF-seeded mice as compared to the PBS-injected group. The demise of nigral TH neurons resulted in the depletion of neurotransmitters in the striatum of PFF-seeded animals (Supplementary Fig. S18d–f). Interestingly, nigral TH neurons, TH protein level, and the striatal DA level were significantly protected in wtNBD-treated mice (Supplementary Fig. S18a–f). As expected, PFF seeding also resulted in a deficit in motor performance of A53T animals as demonstrated by rearing (Supplementary Fig. S18g) and rotarod analysis (Supplementary Fig. S18h). However, concomitant with DAergic neuronal protection, wtNBD treatment significantly inhibited the movement deficits in PFF-seeded mice. These results suggest that intranasal wtNBD peptide was capable of protecting DAergic neurons and improving locomotor activities in PFF-seeded mice.

## Discussion

Prion-like spreading of pathological α-syn aggregates in the brain of α-synucleinopathy such as PD, MSA, and DLB has been well manifested by several studies[6,19,26]. However, mechanisms by which α-syn spreading occurs in the brain are poorly understood. It is well established that seeding of α-syn fibrils in the striatum causes accumulation of pathological form of the protein associated with extensive gliosis in SN that make the tissue

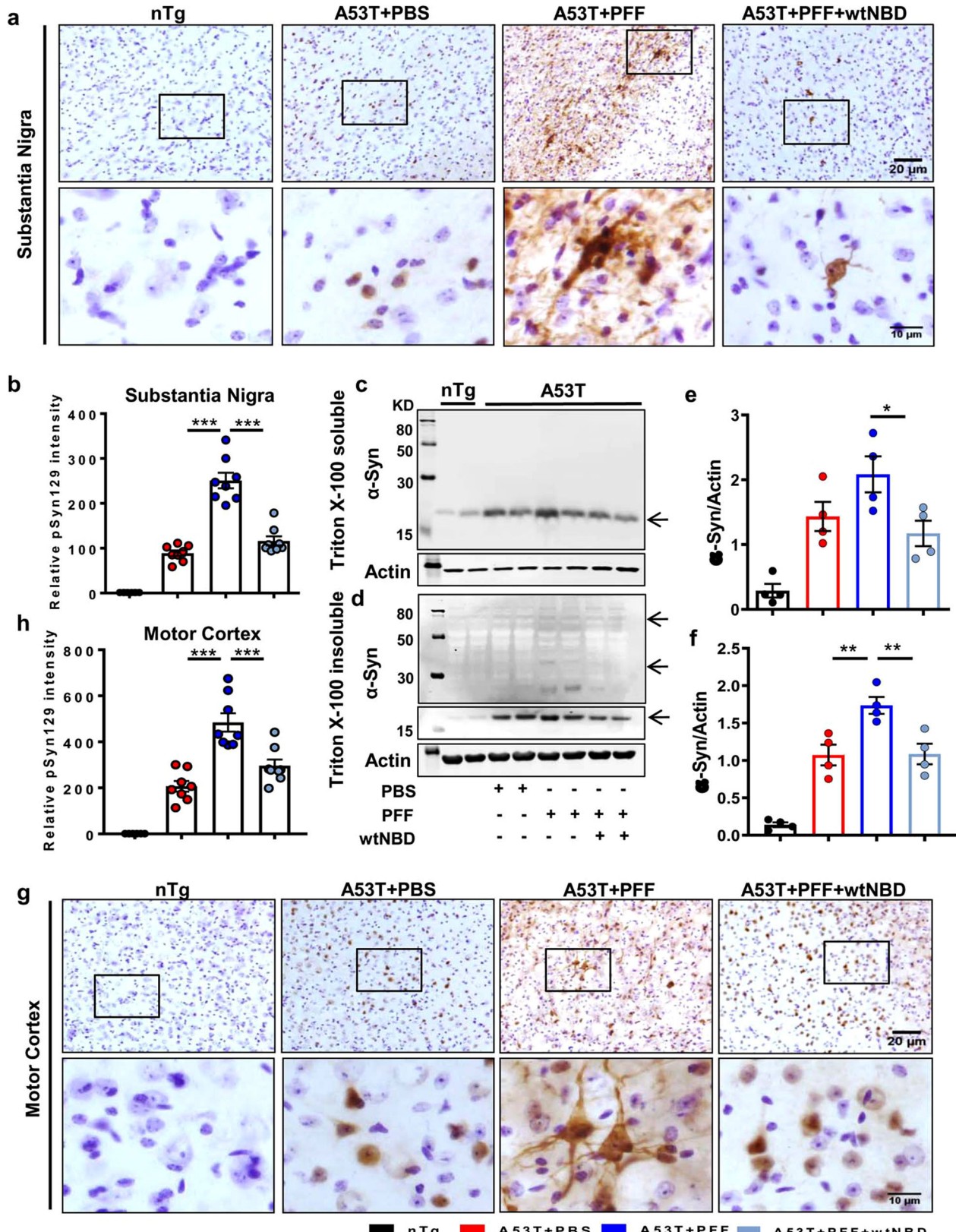

environment hostile for the surviving DAergic neurons[27–29]. An overwhelming level of α-syn leads to the secretion of this protein from neurons[30–33], which then activates microglia via activation of NF-κB activation and upregulation of proinflammatory molecules[34]. In addition, astroglial activation during the course of PD progression happens via reactive microglia-derived factors[16]

and also by the direct influence of neuron-released α-syn, ultimately augmenting the secretion of several proinflammatory factors in the brain[35]. Therefore, aggravated glial activation coincides with increased α-syn expression in the brain. It is also consistent with the previous report that glial activation precedes α-syn pathology in a mouse model of PD[36].

**Fig. 7 Intranasal administration of wtNBD peptide inhibits α-syn spreading in PFF-seeded A53T mice.** Propagation of α-syn in PFF-seeded mice brain was monitored in SN by immunostaining of pSyn129 and relative intensity measurement (**a**, **b**, $p = 0.00001$ for A53T + PBS vs A53T + PFF and $p = 0.00001$ for A53T + PFF vs A53T + PFF + wtNBD). Two sections from each brain were used for the staining and value obtained for each section is plotted in the graph. Spreading was also monitored by immunoblotting of total α-syn in Triton X-100 soluble (**c**, **e**, $p = 0.0426$ for A53T + PFF vs A53T + PFF + wtNBD) and insoluble fractions (**d**, **f**, $p = 0.0067$ for A53T + PBS vs A53T + PFF and $p = 0.0078$ for A53T + PFF vs A53T + PFF + wtNBD). The ratio of α-syn to actin is shown in the diagrams. Level of pSyn129 in the motor cortex was assessed by immunohistochemistry (**g**, **h**, $p = 0.00003$ for A53T + PBS vs A53T + PFF and $p = 0.00017$ for A53T + PFF vs A53T + PFF + wtNBD), where two sections from each brain were used for immunostaining and pSyn129-specific intensity was analyzed by Fiji. One-way ANOVA followed by Tukey's multiple comparison tests was conducted for statistical analyses. **$p < 0.01$, ***$p < 0.001$ indicate significance compared to respective groups. Values are given as mean ± SEM ($n = 4$ animals per group).

Recently, we have delineated that a peptide corresponding to the TLR2-interacting domain of MyD88 (TIDM) specifically inhibits the induction of TLR2 activation in microglia without affecting the basal level of either TLR2 or MyD88[14]. Moreover, the TIDM peptide does not modulate the activation and function of other TLRs[14]. Here, by using TIDM peptide we describe an important role of induced TLR2-MyD88 signaling in microglial activation and α-syn pathology. First, α-syn PFF increased the association between TLR2 and MyD88 in microglia. This physical association in PFF-stimulated microglia was inhibited by wtTIDM, but not mTIDM, peptide. Second, PFF induced the activation of microglia as evident from the activation of NF-κB and the expression of different proinflammatory molecules. However, selective inhibition of TLR2 by wtTIDM led to attenuation of PFF-induced activation of microglia. Third, as expected, widespread microglial activation was seen in the CNS of PFF-seeded mice and aged A53T Tg mice. Intranasal administration of wtTIDM peptide inhibited microglial activation in both cases. Fourth, wtTIDM peptide treatment also reduced the α-syn spreading and pathology in the CNS of PFF-seeded mice and aged A53T Tg mice. Fifth, the specificity of TLR2 inhibition by wtTIDM in vivo was confirmed by conducting the experiments in A53T$^{\Delta TLR2}$ mice, where TLR2 is genetically ablated. A53T$^{\Delta TLR2}$ mice did not respond to wtTIDM treatment in terms of both α-synucleionopathy and glial inflammation, indicating the requirement of functional TLR2 protein for the functioning of wtTIDM peptide. Sixth, PFF seeding in the striatum led to the loss of TH expression and simultaneous induction of apoptotic protein as revealed by the abundance of cleaved caspase 3 in surviving DAergic neurons in the nigra and these facts are likely contributing to the depletion of dopamine in the striatum. However, nasal treatment with wtTIDM peptide led to the protection of nigral TH neurons and striatal dopamine. Seventh, wtTIDM, but not mTIDM, peptide also ameliorated functional impairment in PFF-seeded mice. Because TLR2 level is high in the nigra of post-mortem PD brains[37], our results suggest that wtTIDM peptide may reduce α-synucleinopathy to slow down the loss of nigrostriatal neurons in patients with PD.

According to Kim et al.[7], oligomeric α-syn released from neuron serves as an endogenous agonist of TLR2 to cause paracrine activation of microglia. Accordingly, the non-cell-autonomous neurotoxicity of α-syn is mediated through microglial TLR2[38]. It has been also shown that antagonism of neuronal TLR2 is sufficient for the prevention of α-synucleinopathy[15]. In the absence of any specific inhibitors, these studies used TLR2$^{-/-}$ mice to study the role of TLR2. Others have used TLR2 antagonizing antibodies to block TLR2 in transgenic animal models of α-synucleinopathy[39]. However, these approaches suppress the basal level of the entire TLR2 including its extracellular domain. Therefore, the importance of wtTIDM peptide lies in its efficacy to selectively disrupt the interaction between TLR2 and MyD88 without affecting either its basal level or its interaction with extracellular ligands such as α-syn. Moreover, due to the presence of antennapedia homeodomain, wtTIDM peptide is BBB permeable[14].

It is also indispensable to address that TLR2 was found to be upregulated in astrocytes of the A53T brain. This finding is in line with the earlier finding that showed an increase in astrocytic TLR2 and induction of several inflammatory molecules following exposure to neuron-derived α-syn[10]. On the other hand, fibrillar α-syn-mediated microglial activation is also known to activate astrocytes towards the inflammatory phenotype causing secretion of cytokines like IL-1β, TNF-α, IL-6, etc.[16]. Considering these two points, it can be speculated that astroglial activation may also happen directly in response to α-syn PFF via the TLR2 pathway. Studies have also demonstrated that the expression of antigen-presenting genes is increased in astrocytes upon challenge with α-syn fibrils, indicating the existence of other TLR2-independent pathways in α-syn-exposed astrocytes[40].

How does TLR2 couple α-synucleinopathy? The exaggerated level of insoluble α-syn in multiple brain regions of PFF-seeded A53T mice raised the question of whether only the exogenous α-syn PFF spread in the brain or the PFF induces increased expression of endogenous α-syn to potentiate the seeding phenomenon over time. Since TLR2 is coupled to microglial inflammation, to understand the molecular mechanism in greater details, we tested a hypothesis that proinflammatory molecules might be directly involved in the upregulation of endogenous α-syn. In fact, upregulation of α-syn by a wide variety of proinflammatory molecules including cytokines (IL-1β, TNFα, and IFNγ), bacterial components (LPS and ODN), viral mimics and components (poly IC and Tat), and toxin (MPP$^+$) indicates that inflammatory insults are capable of upregulating α-syn in neurons. The presence of NF-κB binding site in the α-syn gene promoter, recruitment of classical NF-κB heterodimeric components (p65 and p50) and associated p300 HAT to the α-syn gene promoter by IL-1β and TNFα, activation of wild type (pα-syn-WT), but not mutated (pα-syn-Mut), α-syn promoter by various proinflammatory molecules, inhibition of cytokine-mediated upregulation of α-syn by wtNBD peptide, and increase in α-syn by overexpression of NF-κB p65 alone clearly suggest that the upregulation of α-syn is regulated in neurons via classical inflammatory signaling (NF-κB) pathway. Accordingly, intranasal wtNBD peptide inhibited α-syn spreading in the brain and protected dopaminergic neurons in PFF-seeded mice. Here, it is important to mention that Th1 cytokine IFNγ is known to function via STAT, but not NF-κB[41,42]. Accordingly, in contrast to a number of proinflammatory stimuli (IL-1β, TNFα, LPS, ODN, HIV-1 Tat, poly IC, and MPP$^+$), IFNγ induced the activation of α-syn promoter in neuronal cells independent of NF-κB, suggesting that in addition to NF-κB, other inflammatory transcription factor(s) may also be involved in the upregulation of α-syn in neurons.

In summary, we delineated that PFF-induced activation of the TLR2-MyD88 pathway triggered the induction of inflammation and generation of proinflammatory molecules from microglia to further act on neurons and augment endogenous α-syn expression via NF-κB-dependent transcription, ultimately augmenting α-syn pathology in multiple regions of the brain. Accordingly, intranasal administration of wtTIDM (acting at the initial stage to inhibit PFF-induced TLR2 activation) and wtNBD (functioning at

the downstream phase to inhibit TLR2-mediated NF-κB activation) peptides decreased α-syn spreading and protected dopamine in PFF-seeded mice. These results suggest that wtTIDM and wtNBD peptide may be used for therapeutic intervention in PD, MSA, and DLB in which microglial activation and α-synucleinopathy play key roles in disease pathogenesis.

## Methods

**Reagents**. Dulbecco's Modified Eagle Medium (DMEM) medium was purchased from Mediatech (Washington, DC) and fetal bovine serum (FBS) was obtained from Atlas Biologicals (Fort Collins, CO). Antibiotic-antimycotic was purchased from Sigma-Aldrich (St. Louis, MO). Recombinant human α-syn was purchased from Anaspec (Fremont, CA). Rabbit anti-tyrosine hydroxylase (TH) antibody was purchased from Pel-Freeze biologicals (Rogers, AR) and mouse anti-TH antibody was procured from Immunostar (Hudson, WI). Mouse α-syn antibody was purchased from BD Bioscience (San Jose, CA) (Supplementary Table S1). Cy2- and Cy5-conjugated antibodies were obtained from Jackson Immuno-Research Laboratories (West Grove, PA).

**Animals**. Adult C57BL6 mice, TLR2$^{-/-}$ mice (B6.129-Tlr2$^{tm1Kir}$/J) and A53T α-syn transgenic line M83 (B6;C3-Tg(Prnp-SNCA*A53T)83Vle/J) were purchased from Jackson Laboratories. Animal maintenance and experiments were performed in accordance with the National Institutes of Health guidelines and approved by the Institutional Animal Care and Use Committee of the Rush University Medical Center (Chicago, IL). Mice were maintained at room temperatures of 65–75 °F (~18–23 °C) with 40–60% humidity. Mice were kept on a 14/10 h light/dark cycle and given a continuous supply of food and water.

**Preparation and validation of α-Syn fibrils**. For fibril preparation, α-syn monomers were solubilized in 30 mM Tris-HCl (pH 7.4) at the concentration of 350 μM and rotated continuously in a rotary shaker at 250 rpm at 37 °C for 7 days[17,43]. Next, the fibrils were briefly sonicated (15 s) at 10% amplitude and were characterized by performing electron microscopy (EM). For EM imaging, 1 μl of stock solution was diluted in 100 μl of phosphate buffer saline (PBS) and this solution was adsorbed to 300-mesh copper, Formvar-coated EM grid, washed, stained with 2% uranyl acetate, and the grid was allowed to be dried for 15–20 min. Imaging was performed at ×100,000 magnification using a JEOL JEM-1220 transmission electron microscope (operating at 80 kV). Digital micrographs were acquired using an Erlangshen ES1000W model 785 CCD camera and Digital Micrograph software (Version 1.7). The fibril formation was also validated by running the samples in 12% SDS-PAGE followed by immunoblotting using anti-α-syn antibodies (BD Bioscience, San Jose, CA).

**TIDM and NBD peptides**. TIDM and NBD peptides (>99% pure) were synthesized in the custom peptide synthesis facility of Genscript (Piscataway, NJ). TIDM peptides contain the Antennapedia homeodomain (lower case) and six amino acid long MyD88 (upper case) segments[44].

 Wild type (wt) TIDM: drqikiwfqnrrmkwkkPGAHQK
 Mutated (m) TIDM: drqikiwfqnrrmkwkkPGWHQD
 Similarly, NBD peptides also contain the Antennapedia homeodomain (lower case) and six amino acid long IKKβ (upper case) segments[21,45].
 wtNBD:drqikiwfqnrrmkwkkLDWSWL
 mNBD:drqikiwfqnrrmkwkkLDASAL
 Positions of mutations are underlined.

**Mouse BV-2 microglial cells and primary microglia**. BV-2 murine microglial cells (a gift from V. Bocchini, University of Perugia, Perugia, Italy) were maintained in DMEM/F-12 medium containing 10% FBS at 37 °C in the incubator. Primary microglia were isolated from mixed glial cultures as described[21,46,47]. Briefly, brain tissues isolated from 2 to 3 d old mouse pups were dissociated with glass mortar, triturated, passed through the mesh, trypsinized, centrifuged, and mixed glial cells plated in DMEM/F-12 containing 10% fetal bovine serum. On day 9, the mixed glial cultures were washed three times with DMEM/F-12 and subjected to a shake at 240 rpm for 2 h at 37 °C on a rotary shaker to isolate microglia.

**Isolation of primary mouse DAergic neurons**. Similarly, primary DAergic neurons were also isolated as described[48,49]. Briefly, pregnant female mice were euthanized via cervical dislocation to get the embryonic pups (embryonic stage 13.5). Embryonic pups were quickly decapitated in serum-free media to isolate the developing nigra tissue, which was milled three times, and the cells centrifuged at 1000 rpm for 10 min. Pelleted tissue was resuspended in fresh serum-free DMEM, washed three times, and cells plated in 6-well plates containing coverslips. Cells were incubated for 8 days at 37 °C with 5% CO$_2$ in complete DMEM containing 20% FBS and antibiotic-antimycotic prior to their use in subsequent experiments.

**Western blotting**. Western blotting was performed as previously described[50,51]. Equal amounts of proteins were electrophoresed in 10% or 12% SDS-PAGE and transferred onto a nitrocellulose membrane. The blot was probed with primary antibodies overnight at 4 °C (Supplementary Table S1). The next day primary antibodies were removed and the blots were washed with phosphate buffer saline containing 0.1% Tween-20 (PBST) and corresponding infrared fluorophore-tagged secondary antibodies (1:10,000; Jackson Immuno-Research) were added at room temperature (RT). Finally, blots were scanned with an Odyssey infrared scanner (Li-COR, Lincoln, NE). Band intensities were quantified using ImageJ software (NIH, USA).

**Immunoprecipitation (IP)**. BV-2 cells were treated with 10 μM of wtTIDM or mTIDM. After 30 min of TIDM treatment, fibrillar α-syn (0.5 μM or 7 μg/ml) was added to the cells and incubated for 2 h. Following the incubation period, cells were scraped and lysed in RIPA buffer. The cell homogenate was centrifuged at 17,500 × g for 15-min 4 °C, the supernatant was collected and protein was estimated using the BCA method. The cell lysate was immunoprecipitated with 2 μg of anti-TLR2 or anti-MyD88 or normal IgG (Santa Cruz Biotechnology) overnight at 4 °C, followed by incubation with protein A-agarose for 4 h at 4 °C. Protein A-agarose-antigen-antibody complexes were collected by centrifugation at 10,000 × g for 1 min at 4 °C. The pellets were washed 3–4 times with 1 ml of IP buffer containing 20 mM Tris-HCl (pH 8.0), 137 mM NaCl, 2 mM EDTA, 1% Nonidet P-40, 10% glycerol, and protease inhibitor cocktails for 20 min each time at 4 °C. Bound proteins were resolved by SDS-PAGE, followed by western blotting with the anti-MyD88 (1:1000, Santa Cruz Biotechnology) and/or anti-TLR2 (1:1000, Abcam). Input from each sample was also run in the western blotting.

**Chromatin immunoprecipitation (ChIP) assay**. Recruitment of NF-κB to α-syn gene promoters was determined by ChIP assay as described earlier[52]. Briefly, MN9D and SHY5Y cells were treated with IL-1β under serum-free conditions, and after 1 h of stimulation, cells were fixed by adding formaldehyde (1% final concentration), and cross-linked adducts were resuspended and sonicated. In the case of in vivo ChIP, animals were perfused with 4% paraformaldehyde, midbrain was isolated and DNA was extracted from the tissues using the phenol–chloroform–isopropyl alcohol method of DNA isolation. ChIP was performed on the cell lysate by overnight incubation at 4 °C with 2 μg of anti-p65, anti-p50, anti-CBP, anti-RNA Polymerase II, or anti-p300 antibodies followed by incubation with protein G agarose (Santa Cruz Biotechnology) for 2 h. The beads were washed and incubated with elution buffer. To reverse the cross-linking and purify the DNA, precipitates were incubated in a 65 °C incubator overnight and digested with proteinase K. DNA samples were then purified, precipitated, and the precipitate was washed with 75% ethanol, air-dried, and resuspended in TE buffer. Following primers were used for amplification of chromatin fragments of mouse α-syn gene.

 Promoter of α-syn (spanning proximal NF-κB; 237 bp):
 Sense: 5′-TGT ACG CCC ACC TCC CAT GTT CC-3′
 Antisense: 5′-TCA TGT CAC TTA AGG ATG GGA TGG-3′.

**Electrophoretic mobility shift assay (EMSA)**. Nuclear extracts were prepared, and EMSA was performed as described previously[53] with some modifications. Briefly, IRDye end-labeled oligonucleotides containing the consensus binding sequence for NF-κB were purchased from Licor Biosciences. Six micrograms of nuclear extract were incubated with binding buffer and with an infrared-labeled probe for 20 min. Subsequently, samples were separated on a 6% polyacrylamide gel in 0.25× TBE buffer (Tris borate-EDTA) and analyzed by the Odyssey Infrared Imaging System (LI-COR Biosciences).

**Construction of mouse α-syn promoter-driven reporter construct**. Mouse genomic DNA isolated from primary mouse neuron was used as the template during PCR. The 5′-flanking sequence of the mouse α-syn (−1216/+23) gene was isolated by PCR. Primers were designed from GenBank sequences as follows: α-Syn: sense, 5′-acgcgtCCC CCT GCC CCT GCC TGC CCT TG -3′, and antisense, 5′-agatctGTA AGT CCT TTC ATG AAC ACA TC-3′. The sense primer was tagged with a MluI restriction enzyme site, and the antisense primer was tagged with BglII. The PCR was performed using an Advantage-2 PCR kit (Clontech) according to the manufacturer's instructions. The resulting fragments were gel-purified and ligated into the PGEM-TEasy vector (Promega). These fragments were further subcloned into the PGL3 Enhancer vector after digestion with the corresponding restriction enzymes and verification by sequencing (ACGT Inc. DNA Sequencing Services).

**Cloning of α-syn promoter and site-directed mutagenesis**. Site-directed mutagenesis was performed as described earlier[52] by using the site-directed mutagenesis kit (Stratagene). Two primers in opposite orientations were used to amplify the mutated plasmid in a single PCR. The primer sequence for mutated promoter site was as follows: mutated: sense, 5′-GGG AAC TTG ATG GGG TAG AAA ATG TTT ACGTCC CCT TCT G-3′, and antisense, 5′-CAG AAG GGG ACG TAA ACA TTT TCT ACC CCA TCA AGT TCC C-3′. The PCR product was precipitated with ethanol and then phosphorylated by T4 kinase. The phosphorylated fragment was self-ligated by T4 DNA ligase and digested with

restriction enzyme DpnI to eliminate the nonmutated template. The mutated plasmid was cloned and amplified in *Escherichia coli* (DH5-α strain)-competent cells.

**Luciferase assay**. BV-2 cells plated at 50–60% confluence in 12-well plates were transfected with 0.25 µg pNF-κB-Luc construct using Lipofectamine plus (Life Technologies). Following 24 h of transfection, cells were incubated with wtTIDM or mTIDM for 1 h and then exposed to 0.5 µM PFF under serum-free conditions for 2 h. Luciferase activities were analyzed in cell extracts using the Luciferase Assay System kit (Promega) in a TD-20/20 Luminometer (Turner Designs) as described previously[52,54]. Similarly, neuronal cells plated at 50–60% confluence in 12-well plates were transfected with 0.25 µg of either pα-syn-(WT)-Luc or pα-syn-(Mut)-Luc using Lipofectamine Plus. After 24 h of transfection, cells were stimulated with different agents under serum-free conditions for 6 h. Firefly luciferase activities were analyzed in cell extracts.

**ELISA**. ELISA was performed from microglial spent medium to determine the level of secreted IL-1β and TNFα as described before[55]. The experiment was performed using mouse IL-1β and TNFα-specific ELISA kits following the manufacturer's protocol (Invitrogen).

**Phagocytosis Assay**. Primary microglia isolated from WT and TLR2$^{-/-}$ mice were plated either on glass coverslips or in 96-well plates. Cells were treated with 5 µM of wtTIDM or mTIDM for 30 min and then FITC-tagged monomeric α-syn (Anaspec) at 0.25 µM concentration or FITC-tagged latex beads (Cayman chemicals, MI) at 1:1000 dilution was added in each well. Following α-syn addition, cells were kept for 2 h, and phagocytosis of α-syn by the microglia was measured by immunostaining. For LPS-stimulated microglia, LPS was added after 30 min of TIDM treatment for another 1 h and then α-syn was added to the LPS-stimulated microglia to evaluate the phagocytosis. For immunostaining purposes, cells were washed at least three times with a warm medium and then fixed with 4% paraformaldehyde. Then the fixed cells were processed for immunostaining for Iba1 using the procedure described above. Imaging of cells was performed under Olympus BX41 fluorescence microscope and the green fluorescence obtained from each cell of all the groups was measured using ImageJ. MFI values of LPS-treated groups were compared with the respective controls.

**Real-time PCR**. Total RNA was isolated from primary microglia using the Zymogen RNeasy kit following the manufacturer's protocol. The isolated RNA was reverse transcribed into cDNA and real-time PCR was carried out in ABI-Prism7700 sequence detection system (Applied Biosystems, Foster City, CA) using the SYBR green real-time kit obtained from QuantaBio (Beverly, MA) as described before[56,57]. The following primer sequences were used:

iNOS: Sense: 5′-CCCTTCCGAAGTTTCTGGCAGCAGC-3′
Antisense: 5′-GGCTGTCAGAGCCTCGTGGCTTTGG-3′
IL-1β: Sense: 5′-GGA TATGGAGCAACAAGTGG-3′
Antisense: 5′-ATGTACCAGTTGGGGGAACT-3′
GAPDH: Sense: 5′-GGTGAAGGTCGGTGTGAACG-3′
Antisense: 5′-TTGGCTCCACCCTTCAAGGTG-3′.

**Immunostaining**. Animals were perfused with 4% paraformaldehyde, and the brains were kept in a 30% sucrose solution at 4 ℃. Using a cryotome, 30-µm coronal sections were cut from the midbrain region of the brain and processed for immunostaining. In the case of cell samples (primary microglia), these were washed with PBS and then fixed with 4% paraformaldehyde followed by immunostaining. Samples were blocked with either 4% (for tissue sections) or 2% (for cells) BSA in PBS containing 0.5% Tween-20 (Sigma) and 0.05% Triton X-100 (Sigma) for 1 h. Then the samples were kept in primary antibodies for α-syn (1:500, MJFR1, Abcam), psyn129 (1:2000, Abcam), TH (1:1000, Pel-Freeze and Jackson Immunostar), Iba1 (1:1000, Abcam), GFAP (1:2000, Abcam), iNOS (1:200, BD Bioscience) and incubated at 4 ℃ temperature overnight under shaking conditions. The next day, the samples were washed with PBS for 30 min and further incubated with Cy2- or Cy5-labeled secondary antibodies (all 1:500; Jackson Immuno-Research) for 2 h under similar shaking conditions. Following four 15-min washes with PBS, cells were incubated for 4–5 min with 4′,6-diamidino-2-phenylindole (DAPI, 1:10,000; Sigma). For immunohistochemistry of brain sections, samples were kept in a solution containing biotin-tagged secondary antibodies for 2 h followed by incubation in Vectastain A and B mixture solution at RT. Sections were developed by 3,3′-diaminodenzidine (DAB) solution containing peroxide. The sections were mounted on coated slides, dried overnight, and were run in an ethanol and xylene (Fisher) gradient and observed under either Olympus BX41 fluorescence microscope or imaged under Zeiss confocal microscope using Zen 2012 software (Zeiss LSM 780, Carl Zeiss, Jena, Germany). Mean fluorescence intensity (MFI) and counting of target proteins or cells were performed using ImageJ. Intensity of pSyn129 DAB staining was quantified using Fiji (ImageJ2). All figures were deconvoluted to achieve H-DAB stained images and then each cell was outlined and the intensity was measured by the Analyze-measure option provided in the software. The mean value of stained cells in A53T brains (including PBS and PFF-injected) was compared to that of cells of nTg brain. Darker staining means

lower is the mean value. The mean value of white is considered to be the highest (255), and therefore the formula used for calculating relative O.D. is $\log_{10}(255/\text{mean of each cell})$.

**Mouse stereotaxic surgery**. It was performed according to the procedure mentioned by Sorrentino and co-workers[18]. The A53T animals start showing significantly visible α-syn pathology at 4–6 months of stage[20]. Therefore, α-syn PFF was injected into the internal capsule (IC) region of striatum of 3-months-old A53T mice, so that the seeded α-syn fibrils spread up to SN of the midbrain overtime. Animals were injected with 5 µg of α-syn fibrils dissolved in 2.5 µl of PBS in the internal capsule region in both the hemispheres of the brain (coordinates from Bregma: A/P −0.5, L ±1.5, D/V −3.0). Similarly, control animals were injected with only PBS in both the hemispheres. The solution was injected at the rate of 0.4 µl/min and following the injection, the needle was kept for another 5 min on each side of the brain for proper diffusion of the protein solution[18].

**Intranasal treatment of animals with TIDM and NBD peptides**. Wild type (wt) TIDM, mutated (m) TIDM, and wtNBD peptides were solubilized in normal saline in such a way so that each mouse receives 0.1 mg/kg of body weight TIDM/NBD peptide in 2.5 µl of saline. Then 2.5 µl of TIDM/NBD solution was administered in mice through each nostril every day for a total of 30 days. Mice were held in supine condition while administering TIDM/NBD solutions[14]. Intranasal treatment was started for PFF or PBS-injected A53T animals at the age of 5 months (2 months following the brain surgery) for the next 30 days. Aged A53T animals (8 months old) were also treated with TIDM/NBD peptides for 1 month followed by behavioral tests and other experiments at the age of 9 months.

For evaluating the bioavailability of TIDM peptides in mice brain, A53T mice were treated with Alexa-680 tagged wtTIDM or mTIDM through the tail vein. Similarly, control mice were treated with only Alexa-680, which was not tagged with the peptide. Following 2 h of dye injection, animals were scanned under LICOR infrared scanner to visualize the penetration of the peptides in the brain as described before[44]. Lastly, the experimental animals were sacrificed and different blocks of the brain were taken from the cortex, midbrain, and cerebellum region and further scanned under the LICOR scanner.

**Behavioral tests**. Two major kinds of behavioral tests (open field and rotarod) were conducted to evaluate PFF α-syn-induced motor deficits in mice.

*Open field*: Open field test was conducted to look at the locomotor activities of the animals. Locomotor activity was monitored with a camera linked to Noldus system and EthoVisionXT software (Netherlands). The instrument records the overall movement abilities of the animals including total distance moved, velocity, total moving time, resting time, center time, and frequencies of movement and rest. Before the test, mice were placed inside the open field arena for 10 min daily for two consecutive days to train them and record their baseline values. Two days after the training, each mouse was taken from the cage and gently placed in the middle of the open field arena. After releasing the animal, data acquisition was started by the software for the next 5 min and the parameters related to the locomotor activities were collected by the software[58].

*Rotarod*: Prior to the rotarod test, mice were placed on the rotarod instrument for 5–10 min daily for consecutive two days to train them. After 2 days of training, mice were placed on the rotating rod, which rotates with a gradual increasing speed of 4–40 rpm. The experiment was ended if the animals slips from the rotating rod to the base of the instrument or just grips the rod to turn reverse without rotating against the direction of rotating rod[16].

*Pole test*: It was performed with a wooden pole of 57 cm in height and 2 cm in width. Animals were acclimatized for two consecutive days prior to the test and each time five trials were given to each animal. During the test, animal was placed near the top of the pole facing upwards. The time taken by each animal to turn downwards was monitored as the pole turn time. Moreover, the time taken by each animal to reach the base of the pole was recorded as the pole climb downtime. The surface of the pole was made rough by covering it with bandage gauze. The maximum time for recording was set as 60 s. If any animal was found to stall for more than 60 s, the test was further performed for that particular animal[16,59].

**Tissue lysate preparation**. Midbrain tissues were isolated from experimental mice and homogenized in Triton X-100 soluble buffer containing 1% Triton X-100, 0.5 µM EDTA, 10 mM Tris-HCl (pH 7.4), 150 mM NaCl and protease inhibitors, and phosphatase inhibitor cocktails. The homogenate was centrifuged at 17,500 × *g* for 15 min at 4 ℃. The supernatant obtained after centrifugation was the detergent soluble fraction of the tissue. The resultant pellet was dissolved in detergent-insoluble buffer containing 50 mM Tris, pH 8.0, 1% Triton X-100, 2% SDS, 1% sodium-deoxycholate, 1% NP-40, 0.5 µM EDTA and protease, phosphatase inhibitor cocktails. The lysate was further centrifuged at 17,500 × *g* for 15 min at 4 ℃ and the supernatant was collected as the Triton X-100 insoluble fraction[16,43]. Total tissue lysate was obtained by dissolving the tissue in RIPA buffer (50 mM Tris, pH 8.0, 150 mM NaCl, 1% Nonidet P-40, 1% SDS, 0.5% sodium-deoxycholate) with complete protease and phosphatase inhibitor cocktails. Tissues were sonicated for 20–30 s and the homogenate was centrifuged at 17,500 × *g* for 15 min at 4 ℃ and the resulting supernatant was collected[50].

**TH neuronal counting**. TH immunohistochemistry in both SN and striatum was performed according to the protocol mentioned earlier[58,60]. Counting of TH neurons in SN of each hemisphere of the brain was performed by using STEREO INVESTIGATOR software (MicroBrightfield, Williston, VT) having an optical fractionator as described before[21,60]. TH fiber density in the striatum was measured by using the Fiji software.

**HPLC analysis for measurement of striatal DA and its metabolite levels**. Striatal levels of DA, DOPAC, and HVA were measured as mentioned earlier[58]. In brief, after 7 days of MPTP treatment, mice were sacrificed by cervical dislocation, and striatum was isolated from each mouse and sonicated in 0.2 M ice-cold perchloric acid. The homogenate was centrifuged at $17,500 \times g$ for 15 min at 4 °C. Supernatant was collected in a fresh tube and 10 µl of the supernatant was injected in an Eicompak SC-3ODS column (Complete Stand-Alone HPLC-ECD System EiCOMHTEC-500; JM Science, Grand Island, NY). Levels of neurotransmitters were analyzed according to the manufacturer's protocol.

**Statistics**. Statistics were performed using GraphPad Prism v7.0. Values are expressed as mean ± S.D. for data obtained from cellular studies and mean ± S.E. for animal experiments. At least three experiments were conducted for each analysis. Statistical analyses for differences between two different samples were performed using an unpaired two-tailed $t$-test. One-way ANOVA followed by Tukey's multiple comparisons was performed for statistical analyses among multiple groups. Two-way ANOVA was used for comparing more than one parameter among different groups. The criterion for statistical significance was $p < 0.05$.

**Reporting summary**. Further information on research design is available in the Nature Research Reporting Summary linked to this article.

## Data availability

All data supporting the findings of this study are provided within the paper and its Supplementary Information. A source data file is provided with this paper. All additional information will be made available upon reasonable request to the authors. Source data are provided with this paper.

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

## Acknowledgements

This study was supported by grants from the National Institutes of Health (NS108025 and AG069229). Moreover, Dr. Pahan is the recipient of a Research Career Scientist Award (1IK6 BX004982) from the Department of Veterans Affairs.

## Author contributions

D.D., M.J. and K.P. designed the study. D.D., M.J., M.M., S.M. and A.R. conducted the experiments. D.D., M.J. and K.P. analyzed and interpreted the data. D.D. and K.P. wrote the manuscript. K.P. conceived the idea and received funding.

## Competing interests

The authors declare no competing interests.
