## [Peer Review File · Nature Communications]

Reviewers' Comments:

Reviewer #1:

Remarks to the Author:

The manuscript by Dutta D. et al. 2020 entitled "Reduction of α -synuclein spreading by selective targeting of the TLR2/MyD88/NF- κ B pathway" investigates how TLR2 and NF- κ B-mediated inflammatory pathways affect the spreading of alpha-synuclein (α -syn) and neurodegeneration related to PD. The authors provide evidence that intranasal administration of the peptides TIDM (to block TLR2 signaling) or NEMO-binding domain (to block NF- κ B activation) reduced α -syn spreading, decreased reactive gliosis, attenuated neurodegenerative pathology, and restored locomotor activity in mice associated with PD. The authors further demonstrate in vitro that pro-inflammatory cytokines induce α -syn production in neurons via NF- κ B and NF- κ B binds directly to the α -syn promoter to regulate its expression in neurons. This manuscript provides important new mechanistic insight into how neuroinflammation drives spreading neurodegenerative pathology in PD and other diseases involving α -syn. The combination of elegant pharmacological in vivo strategies to manipulate these inflammatory pathways with in vitro studies to more carefully address mechanism are powerful. Concerns are outlined below:

Major concerns:

1. The authors do not provide sufficient evidence to support the conclusions that inflammatory cytokines via TLR2 signaling in microglia in vivo modulates α -syn spreading. Inhibitors used in vivo are not cell-specific. Without better evidence of cell specificity in vivo, conclusions related to cell-specificity and microglia should be toned down.
2. The authors have not convincingly demonstrated that TLR2 activation in microglia is upstream of NF- κ B mediated transcriptional regulation of α -syn in neurons. Instead, they show that global TLR2 inhibition protects from α -syn spreading and neurodegeneration. They also show that administration of inflammatory cytokines, that can be downstream of TLR2 but also many other receptors, regulates the transcriptional regulation of α -syn in cultured neurons through NF- κ B. To more convincingly show that NF- κ B dependent transcriptional regulation of α -syn is TLR2-dependent, the authors would need to show in either a neuron-microglia co-culture system or in vivo (e.g. A53T+PFF+mTIDM in vivo) that TLR2 blockade prevents NF- κ B transcriptional regulation of α -syn in neurons. This would still not address whether this effect is microglia-specific, but it would address the TLR2-dependent mechanism.
3. Similar to the point above, the first half of the manuscript assessing TLR2-deficiency effects in vitro and in vivo and the second half of the manuscript assessing NF- κ B regulation of α -syn transcription in neurons are a bit disjointed. As written, it reads like two distinct stories. Incorporating TLR2 signaling into the last half of the paper would help.
4. Measurements of reactive gliosis are rather crude and could use more extensive characterization. The anti-Iba-1 marker is not specific to microglia. It will also label other myeloid-derived cells that are either resident to the CNS (e.g. perivascular macrophages) or infiltrating due to BBB breakdown. More specific markers for homeostatic microglia such as P2RY12 and DAM/MgND microglial such as Clec7a should be used (see Krasemann et al. Immunity 2017 or Keren-Shaul Cell 2017). Similar is true for astrocytes where GFAP is not the best marker of reactive astrogliosis (see Liddelow et al. Nature 2017) and a more global marker of astrocytes such as ALDH1 would be helpful.
5. The manuscript would greatly benefit from clearer descriptions of the mouse models and the reasoning for the experimental paradigms (e.g. A53T mouse model, PFF strategy, time points chosen, etc.).
6. Do younger A53T animals without PBS or PFF injection have increased Iba1, iNOS, Il1b at baseline? Otherwise, it appears that sham injection (PBS alone) causes an inflammatory response and the authors should comment on this in the results.

Minor concerns:

1. The authors refer to BV2 cells as microglia making it difficult to follow the experimental paradigms between primary microglia and cell lines tested. As BV2 cells have a molecular profile that is even further from in vivo microglia than primary microglia cultures (e.g. Das et al. J. Neuroinflammation 2016), it's most appropriate to refer to them distinctly as BV2 cells and primary microglia.
2. There are several typos that require attention (e.g. Iba1+ ve cells).
3. In Fig 1, the authors mentioned the use of standard deviation to report values and SEM for the other figures. For n's ≥ 3 , SEM is most appropriate.

4. Fig 1B and 1C show discrepancies in the concentrations used to validate PFF formation (1-4ug) and treatment in cells (0.5ug). The authors should state the reasoning for using a lower concentration in the text.
5. In Supplemental Figure 2F, include Iba1 and iNOS blots containing the lower band from the last two lanes or indicate which band is quantified from the samples.
6. All western blots for soluble and insoluble conditions should include arrows indicating the bands corresponding to α -syn.
7. In Figure 4A,B showing the number of TH neurons, how were the number of cells quantified and normalized (# cells/ μ m²)?
8. In Supplemental Figure 4A, is α -syn (pSyn129) detectable in A53T mice at baseline?
9. The image of the A53T+PFF+mTIDM condition in Fig S4 hippocampus appears to be at a different magnification.
10. How was normalization per animal per group performed for behavior and histology? This was not clear in the methods.
11. It is unclear what sex of animals was used for these studies.

Reviewer #2:

Remarks to the Author:

The manuscript by Dutta et al. shows that alpha-synuclein preformed fibrils (PFF) activate TLR2-MyD88 in microglia and that an antagonizing peptide for the TLR2-MyD88 interaction can rescue the animals from microglial activation, alpha-synuclein spreading, dopaminergic degeneration, and motor behavioral deficits in the PFF injection model. The authors then show that inflammatory factors induce the expression of alpha-synuclein in neurons through the activation of NF- κ B. Furthermore, an inhibitory peptide for NF- κ B has the similar protective effects as the TLR2-Myd88 blocking peptide. Although the therapeutic implications of using peptides for synucleinopathies are interesting, the role of TLR2 and its signaling in alpha-synuclein-induced inflammation and spreading has been reported previously. More than half the data presented in this manuscript is confirmation of the previous studies. Novel aspect of the manuscript lies in the set of data related to the involvement of neuronal NF- κ B in alpha-synuclein gene expression.

Specific comments:

Results, the first sentence: It is inappropriate to call sonicated fibrils protofibrils.

Fig. 1C and D: These data are critical and thus, should be confirmed in primary microglia.

Fig. 1 K-M: Here, internalization of FITC-tagged monomeric alpha-synuclein was analyzed for phagocytosis assay. However, the internalization of monomeric alpha-synuclein is unlikely to occur through phagocytosis, making it an inappropriate assay for phagocytosis. I suggest the authors to perform the standard assay for phagocytosis, such as internalization of labeled beads or bacterial components.

Fig. 2A-D: Please comment on why the high molecular weight species were not reduced in the wtTIDM-treated animals, while other data showed the decrease in the HMW alpha-synuclein (e.g., Fig. 2J and Fig. 3C).

Page 10, "Finally, motor performance of these animals...": This is an over-interpretation, since it is not clear whether these behavioral phenotypes were derived from the dopaminergic degeneration.

Fig. 6: Evidence should be provided to support that the wtNBD inhibited the 'neuronal' NF- κ B to reduce the alpha-synuclein expression and pathology.

Page 17, "Therefore, glial activation has an important role to play in α -syn spreading.": No direct evidence for this statement was provided.

Reviewer #3:

Remarks to the Author:

The current study leverages the wtTIDM (TLR2-interacting domain of MyD88 peptide), which was previously developed in the author's lab, to treatment various synucleinopathy mouse models and reports the attenuation of PFF-induced microglial inflammation, decrease in alpha-synuclein spreading, and neuroprotection. The study also shows that a few proinflammatory cytokines increase the transcription of alpha-synuclein gene in neurons through NF-kB activation, which is inhibited by NEMO-binding domain peptide (wtNBD). While the notion of TLR2-MyD88-NF-kB in the regulation of alpha-synuclein-associated neuroinflammation and spreading has previously been shown, the increase of alpha-synuclein expression induced by proinflammatory molecules in neurons is interesting and novel. However, there are multiple aspects of the reports that need additional investigations in order to support the main conclusions.

The main concerns are the following:

1. For both peptides that are administrated in mice, the bioavailability and target engagement are poorly or not characterized, therefore, the action of the peptides is unclear and the results are not convincing. For example, what is the evidence for the wtTIDM penetration of BBB and disruption of TLR2-MyD88 interaction in the mouse brains?
2. It is difficult to understand what the exact role TLR2 receptor plays for PFF-synuclein phagocytosis. On the one hand, TLR2 is activated with PFF treatment; on the other hand, disruption of TLR2-MyD88 interaction does not alter the PFF phagocytosis. Have the authors tried chemical inhibitors that block phagocytosis to validate the conclusion of unaltered phagocytosis?
3. The authors used MN9D and SH-SY5Y cell lines only to test the effect of proinflammatory cytokines in increasing alpha-synuclein expression. However, this is not entirely convincing for the conclusion as they are not real neurons. The authors are recommended to use at least primary neuronal cultures to verify the conclusions. Furthermore, the authors failed to see any increase of alpha-synuclein in vivo (Fig. 2C, 2K, 3D). Therefore, it is a direct conflict with the claim of increased expression of alpha-synuclein by proinflammatory cytokines.

Additional concerns/comments:

1. In Immunoprecipitation data (Fig. 1C), the authors should perform the IgG IP and Myd88 IP at the same time. Also, they need to provide immunoblotting for PFF detection in Input section.
2. To test the effect of TLR2 on synuclein uptake, the authors used GFP-tagged monomeric synuclein, which is irrelevant to PFF. The authors may use GFP-tagged PFF instead. This problem applies to Fig. 1I, J, K, and L.
3. In Fig 2B, the authors cut the blot between 15 and 30 kda to show monomeric and high-molecular weight of alpha-synuclein, separately. This is not convincing. It is highly recommended that the authors provide the whole gel as shown in Fig 2J.
4. There is a lack of detailed information regarding the imaging quantification. For example, it is unclear how the authors quantified p-S129 synuclein signal in Fig 2F,2H,3G,3I. Did the authors quantify the signals per cell? What does each dot in the graph indicate? How did the authors exclude the decrease of pS129-synuclein signal from the death of dopaminergic neurons, which occurs in PFF-model? Besides, images of A53T+PFF+mTIDM condition shows intense p-S129 synuclein signal (Fig 2E, G), but does not match the quantification graph.
5. The result showed more pS129-synuclein signals (Fig. 2E) in A53T+PFF group where dopaminergic neurons are dying (Fig. 4A). This is inconsistent with the previously published data (Luk et al., 2016, Science), which showed a reduction of pS129-synuclein-positive cells due to the death of dopaminergic neurons. The authors should clarify this.
6. It is recommended to use "µg/ml" unit, not "µM", to describe the concentration of PFF.
7. In Fig 4, the claim for the loss of DA neurons based on counting of TH+ cells could be misleading. It is recommended to include Nissl staining to assure the loss of DA neurons rather than reduction of TH expression.

Point-by-point response to referees:

Revision of the manuscript (NCOMMS-20-23633-T) entitled “Reduction of α -synuclein spreading by selective targeting of the TLR2/MyD88/NF- κ B pathway”

Reviewer #1:

The manuscript by Dutta D. et al. 2020 entitled “Reduction of α -synuclein spreading by selective targeting of the TLR2/MyD88/NF- κ B pathway” investigates how TLR2 and NF- κ B-mediated inflammatory pathways affect the spreading of alpha-synuclein (α -syn) and neurodegeneration related to PD. The authors provide evidence that intranasal administration of the peptides TIDM (to block TLR2 signaling) or NEMO-binding domain (to block NF- κ B activation) reduced α -syn spreading, decreased reactive gliosis, attenuated neurodegenerative pathology, and restored locomotor activity in mice associated with PD. The authors further demonstrate in vitro that pro-inflammatory cytokines induce α -syn production in neurons via NF- κ B and NF- κ B binds directly to the α -syn promoter to regulate its expression in neurons. This manuscript provides important new mechanistic insight into how neuroinflammation drives spreading neurodegenerative pathology in PD and other diseases involving α -syn. The combination of elegant pharmacological in vivo strategies to manipulate these inflammatory pathways with in vitro studies to more carefully address mechanism are powerful.

Response: Thanks for enthusiastic comments.

Major concerns:

1. The authors do not provide sufficient evidence to support the conclusions that inflammatory cytokines via TLR2 signaling in microglia in vivo modulates α -syn spreading. Inhibitors used in vivo are not cell-specific. Without better evidence of cell specificity in vivo, conclusions related to cell-specificity and microglia should be toned down.

Response: We are thankful to the reviewer for providing such an important suggestion. It is obviously true that neither any cell specific TLR2 inhibitors nor any cell specific TLR2 knockout mice have been used in the study. However, when A53T^{TLR2} animals were compared with only A53T animals, in both sporadic (Figure 3) and genetic (Figure S9) models, we observed less α -syn spreading and reduced accumulation of α -syn in neurons respectively and that correlated with attenuated microgliosis and inflammation (Figure S9). Moreover, in the co-culture studies, we have shown that when microglial TLR2 is inhibited by wtTIDM, microglia is unable to produce exaggerated amount of PFF-induced inflammatory mediators (Figure S7), which results in attenuated α -syn up-regulation in primary neurons. Collectively, these observations firmly indicate that inhibition of microglial TLR2, if not solely, but to a great extent is involved in PFF-induced microglial inflammation and α -syn up-regulation in neurons.

2. The authors have not convincingly demonstrated that TLR2 activation in microglia is upstream of NF- κ B mediated transcriptional regulation of α -syn in neurons. Instead, they show that global TLR2 inhibition protects from α -syn spreading and neurodegeneration. They also show that administration of inflammatory cytokines, that can be downstream of TLR2 but also many other receptors, regulates the transcriptional regulation of α -syn in cultured neurons through NF- κ B. To more convincingly show that NF- κ B dependent transcriptional regulation of α -syn is TLR2-dependent, the authors would need to show in either a neuron-microglia co-culture system or in vivo (e.g. A53T+PFF+mTIDM in vivo) that TLR2 blockade prevents NF- κ B

transcriptional regulation of α -syn in neurons. This would still not address whether this effect is microglia-specific, but it would address the TLR2-dependent mechanism.

Response: We are obliged to the reviewer for making another important comment and that helped us make the study more robust. To address the point that NF- κ B dependent transcriptional regulation of α -syn in neurons is microglial TLR2-dependent, we have performed co-culture experiments where microglia was kept in the inserts and the primary dopaminergic neurons were cultured in the bottom well. The experimental design and outcome are shown in Figure 6A and Figure S14C. In the co-culture experiment, we have shown that when microglial TLR2 is inhibited specifically by wtTIDM, but not mTIDM, the generation of microglia-derived inflammatory cytokines is reduced and the activation of NF- κ B is decreased, leading to attenuated α -syn expression in neurons following co-culturing with PFF-treated microglia. In another set of experiments, we have demonstrated when neuronal NF- κ B is inhibited specifically by wtNBD, but not mNBD, prior to co-culturing with PFF-treated microglia, it also results in reduced expression of α -syn in neurons. In addition to that, we have conducted immunostaining of Ser536 phospho-p65, the RelA subunit of NF- κ B, in nigral dopaminergic neurons (Fig. 6), which shows reduced expression of this protein in wtTIDM-treated mice compared to PFF-treated group. Along with that the ChIP analysis shows reduced recruitment of NF- κ B subunits, RNA polymerase II and transcriptional co-activator p300 to α -syn promoter in wtTIDM-treated PFF-seeded mouse brains (Fig. 6H-I), indicating that TLR2 inhibition markedly reduces NF- κ B mediated transcriptional activation of α -syn.

3. Similar to the point above, the first half of the manuscript assessing TLR2-deficiency effects in vitro and in vivo and the second half of the manuscript assessing NF- κ B regulation of α -syn transcription in neurons are a bit disjointed. As written, it reads like two distinct stories. Incorporating TLR2 signaling into the last half of the paper would help.

Response: Here, we have demonstrated that PFF induces TLR2-MyD88 interaction in microglia to generate induction of inflammation via NF- κ B (Figure 1C-1G) and when this interaction is inhibited by wtTIDM peptide, microglial inflammation is greatly decreased. It clearly suggests that TLR2 activation is much upstream to the activation of NF- κ B in microglia. However, involvement of NF- κ B activation in neurons comes into play when we addressed the transcriptional regulation of α -syn in neurons. This perhaps is the first report which shows the mechanism behind α -syn up-regulation in the context of neuroinflammation. As this cascade of pathways is initiated from the activation of microglial TLR2, therefore we initiated our investigation by addressing TLR2-mediated microglial inflammation and its resultant effect on α -syn pathology, then we moved on to targeting NF- κ B-induced α -syn expression in neurons with NBD peptides.

4. Measurements of reactive gliosis are rather crude and could use more extensive characterization. The anti-Iba-1 marker is not specific to microglia. It will also label other myeloid-derived cells that are either resident to the CNS (e.g. perivascular macrophages) or infiltrating due to BBB breakdown. More specific markers for homeostatic microglia such as P2RY12 and DAM/MgND microglial such as Clec7a should be used (see Krasemann et al. Immunity 2017 or Keren-Shaul Cell 2017). Similar is true for astrocytes where GFAP is not the best marker of reactive astrogliosis (see Liddelow et al. Nature 2017) and a more global marker of astrocytes such as ALDH1 would be helpful.

Response: We further thank the reviewer for the valuable suggestion. Accordingly we tried to evaluate astrogliosis by using ALDH1. However, as the region of interest was substantia nigra, we found some of the dopaminergic neurons were also stained for ALDH1, and that is supported by several other

studies which have shown that nigral dopaminergic neurons also do express ALDH1, which function to produce GABA by these neurons (Liu et al., 2014; Wu et al., 2019). Therefore, by ALDH1 staining it is difficult to solely stain astrocytes, which were perfectly stained by GFAP as shown by previous studies also (Sorrentino et al., 2017; Yun et al., 2018). Moreover, co-staining of GFAP and iNOS further proved the induction of inflammation and stress by resident astrocytes.

On the other hand, in case of microglial staining the suggested marker like P2RY12 or Clec7a could have been used, but the mentioned article by Krasemann et al., 2017 have demonstrated that the expression of these markers change according to the disease state in AD mouse model, for example the homeostatic marker P2RY12 expression decreases variably when there is A β pathogenesis because P2RY12⁺ microglia were preserved in A β diffuse plaques, whereas P2RY12 signal was lost in neuritic plaques. In contrast Clec7a is up-regulated in disease associated microglia. Therefore, these markers might be better used in the context of plaque pathology, and yet less explored in case of synucleinopathy or Parkinson's disease. Moreover, another mentioned study by Keren-Shaul et al., 2017 also used Iba1 as a primary marker of microglia in animal and human brains. This report along with several other reports (Kim et al., 2013; Yun et al., 2018) supports our present experimental approach to identify microgliosis by Iba1 staining.

5. The manuscript would greatly benefit from clearer descriptions of the mouse models and the reasoning for the experimental paradigms (e.g. A53T mouse model, PFF strategy, time points chosen, etc.).

Response: The description for choosing the animal models as well as the experimental paradigm has been provided in the results section under the subsection 'Intranasal administration of wtTIDM peptide reduces gliosis in the SN of PFF-seeded mice' and 'The wtTIDM peptide prevents α -syn-induced pathology in aged A53T model of α -synucleinopathy'.

6. Do younger A53T animals without PBS or PFF injection have increased Iba1, iNOS, I11b at baseline? Otherwise, it appears that sham injection (PBS alone) causes an inflammatory response and the authors should comment on this in the results.

Response: At the end of 6 months of age both the PBS and PFF-injected animals were sacrificed in the present study. At this time point, certain level of inflammation is found in normal A53T+PBS animals and that is increased significantly by PFF. Microgliosis is observed in SN of A53T animals even before 6 months of age as shown by Li et al., 2019 (Li et al., 2019). Therefore, in the sham injected A53T mice as well, we found inflammation, which is comparable to only A53T animals but higher than nTg animals.

Minor concerns:

1. The authors refer to BV2 cells as microglia making it difficult to follow the experimental paradigms between primary microglia and cell lines tested. As BV2 cells have a molecular profile that is even further from in vivo microglia than primary microglia cultures (e.g. Das et al. J. Neuroinflammation 2016), it's most appropriate to refer to them distinctly as BV2 cells and primary microglia.

Response: As suggested by the reviewer, we have distinctly mentioned BV-2 cells wherever needed in the results section under the subsection 'Inhibition of PFF-induced TLR2 activation by wtTIDM peptide'.

2. There are several typos that require attention (e.g. Iba1+ ve cells).

Response: The manuscript has been thoroughly edited to take care of the typos.

3. In Fig 1, the authors mentioned the use of standard deviation to report values and SEM for the other figures. For n's ≥ 3 , SEM is most appropriate.

Response: In the entire study, we have represented the data obtained from cell culture experiments as mean \pm SD (n=3), whereas results obtained from *in vivo* experiments are presented as mean \pm SEM (n>3).

4. Fig 1B and 1C show discrepancies in the concentrations used to validate PFF formation (1-4ug) and treatment in cells (0.5ug). The authors should state the reasoning for using a lower concentration in the text.

Response: We performed Western blot analysis for the PFF at concentrations 1-4 μ g to visualize the increase in higher molecular weight α -syn species. However, while performing the experiments with primary microglia, we initially did at varying concentrations starting from 0.5-2 μ M and found remarkable increase in inflammatory molecules even at the lowest dose 0.5 μ M (7 μ g/ml). We also reviewed the existing reports on PFF concentration on primary cells and based on that for further experiments we continued with 0.5 μ M PFF.

5. In Supplemental Figure 2F, include Iba1 and iNOS blots containing the lower band from the last two lanes or indicate which band is quantified from the samples.

Response: In case of both Iba1 and iNOS bands, the upper bands have been measured for intensity analysis because these bands matched or were at more proximity to the given molecular weight of these proteins (Iba1 is 17 KD and iNOS is 130 KD).

6. All western blots for soluble and insoluble conditions should include arrows indicating the bands corresponding to α -syn.

Response: We have marked the band of α -syn monomer in both soluble and insoluble fractions. In soluble fraction we obtained only α -syn monomer band, but in insoluble fraction along with α -syn monomer band several higher molecular weight band of oligomeric α -syn also appeared and those were also marked with arrows, although the presence of these bands differed from samples to samples in some experiments. Therefore, for measuring band intensity only the bands of α -syn monomer have been considered except Figure 5, where α -syn expression was measured from MN9D cells.

7. In Figure 4A,B showing the number of TH neurons, how were the number of cells quantified and normalized (# cells/ μ m²)?

Response: The number of TH neurons was counted from one hemisphere of each mouse brain using stereology apparatus where every 6th section was counted and the section width was 30 μ m. Based on the final counting of 6 sections from each brain, the software provides the final number of total TH neurons in each hemisphere of the brain. Therefore, the counting has not been represented as cells/mm², but as the total number of TH neurons in one hemisphere of a brain.

8. In Supplemental Figure 4A, is α -syn (pSyn129) detectable in A53T mice at baseline?

Response: As mentioned earlier, A53T mice exert α -syn pathology following 4 months of age (Brahmachari et al., 2016), therefore at the age of 6 months these mice definitely exhibit pSyn129 staining in hippocampus, which is quite evident from the picture in Figure S7A.

9. The image of the A53T+PFF+mTIDM condition in Fig S4 hippocampus appears to be at a different magnification.

Response: The image for the A53T+PFF+mTIDM in hippocampus has been taken at the identical magnification as for other groups. For further accuracy, images have been newly arranged for this figure. This figure is now named as Figure S7.

10. How was normalization per animal per group performed for behavior and histology? This was not clear in the methods.

Response: For behavioral analyses, all experimental mice were acclimatized to certain behavioral environments and to the instrumental set up. The first two days animals were trained for each behavior and then following a gap of one day the final tests were performed. The training of animals was also carried out to deduct the baseline activities of experimental animals. In all the behavioral assays, at least 6 mice were used per group and the individual values obtained from each animal were considered for statistical analyses.

In case of histology, 2 sections from each brain from all the groups were collected and stained for a particular protein under the same conditions including the washes, permeabilization, antibody dilutions and developing the staining signal. The images were captured under the microscope keeping the microscope setting, exposure and magnification same for all the groups for imaging a particular target protein. The mean fluorescent intensity or optical density of the target protein in a particular brain region (for example substantia nigra or motor cortex) was measured from at least two different fields of every section. Thereby, two individual intensity values were obtained from each brain (2 sections/brain) for any given staining and these values were considered for calculating the mean value of that staining for a specific group.

11. It is unclear what sex of animals was used for these studies.

Response: In the present study, both male and female animals have been used for both the PFF-induced sporadic model and for the aged A53T genetic model.

Reviewer #2:

The manuscript by Dutta et al. shows that alpha-synuclein preformed fibrils (PFF) activate TLR2-MyD88 in microglia and that an antagonizing peptide for the TLR2-MyD88 interaction can rescue the animals from microglial activation, alpha-synuclein spreading, dopaminergic degeneration, and motor behavioral deficits in the PFF injection model. The authors then show that inflammatory factors induce the expression of alpha-synuclein in neurons through the activation of NF-kB. Furthermore, an inhibitory peptide for NF-kB has the similar protective effects as the TLR2-Myd88 blocking peptide. Although the therapeutic implications of using peptides for synucleinopathies are interesting, the role of TLR2 and its signaling in alpha-synuclein-induced inflammation and spreading has been reported previously. More than half the data presented in this manuscript is confirmation of the previous studies. Novel aspect of the manuscript lies in the set of data related to the involvement of neuronal NF-kB in alpha-synuclein gene expression.

Response: Thanks for nice comments.

Specific comments:

Results, the first sentence: It is inappropriate to call sonicated fibrils protofibrils.

Response: We have changed the protofibrils to short length fibrils.

Fig. 1C and D: These data are critical and thus, should be confirmed in primary microglia.

Response: Although we have used primary mouse microglia for different experiments (Fig. 1G-L & Fig. S14), in order to perform immunoprecipitation-coupled immunoblot analysis, we would require much more primary microglia at the same time. Therefore, we have performed this experiment in BV-2 cells (Fig. 1C-D). We have also repeated this experiment in BV-2 cells to run all anti-MyD88-pulled down samples and IgG samples in a same gel (Fig. S1).

Fig. 1 K-M: Here, internalization of FITC-tagged monomeric alpha-synuclein was analyzed for phagocytosis assay. However, the internalization of monomeric alpha-synuclein is unlikely to occur through phagocytosis, making it an inappropriate assay for phagocytosis. I suggest the authors to perform the standard assay for phagocytosis, such as internalization of labeled beads or bacterial components.

Response: Thank you for such an important comment. However, previous reports have demonstrated that microglia can phagocytose monomeric α -syn and that its phagocytic efficiency is reduced in case of aggregated α -syn (Park et al., 2008). Keeping that point in consideration, we have used FITC-tagged α -syn monomers to evaluate normal phagocytosis by primary microglia. However, as suggested by the reviewer we have also performed the assay using FITC-tagged latex beads and that is presented in Figure S2.

Fig. 2A-D: Please comment on why the high molecular weight species were not reduced in the wtTIDM-treated animals, while other data showed the decrease in the HMW alpha-synuclein (e.g., Fig. 2J and Fig. 3C).

Response: It was an important point made by the reviewer. We have repeated the experiment in different set of mice and found that high molecular weight species of α -syn in the insoluble fraction are also markedly decreased in SN of wtTIDM-treated animals similar to the finding obtained from cortical tissues. Therefore, a new representative blot is provided for the Figure 2B.

Page 10, “Finally, motor performance of these animals....”: This is an over-interpretation, since it is not clear whether these behavioral phenotypes were derived from the dopaminergic degeneration.

Response: In the experimental PFF-injected mice, previous reports have confirmed loss of nigral TH neurons and depleted dopamine in striatum at varying time points following PFF injection. However, majority of these studies were conducted either in wild type mice (Luk et al., 2012; Rey et al., 2019) or M20 mice (Sorrentino et al., 2017; Yun et al., 2018). In contrast, when sonicated PFF fibrils were injected in M83+/- mice, it resulted in hind limb paralysis in mice as shown by Lau et al., 2020 (Lau et al., 2020). Similarly, in our study, we have used M83+/+ mice carrying only mutated form of α -syn and following 3 months of PFF injection we observed significant demise of TH neurons and loss of striatal dopamine, which correlated well with reduced horizontal movement in open field analysis and also with reduced motor coordination. In addition, we also found tumbling behavior for some of the PFF-injected mice, therefore those mice were unable make proper use of hind limbs even in horizontal motor

activities. We also attempted to perform pole test for these animals, however, some of the PFF-injected animals were unable to climb down to the base of the pole using the grips and were falling down suddenly from the pole tip. This observation restricted us from including pole test analysis in the behavioral data. We could expect to perceive even more pronounced TH neuronal death and behavioral abnormalities, if these mice are kept for longer time like up to 6 months following surgery. To achieve robustness in the data, we carried out the behavioral performance in at least six mice in each group and found significant difference between PBS-injected and PFF-injected groups.

Fig. 6: Evidence should be provided to support that the wtNBD inhibited the 'neuronal' NF- κ B to reduce the alpha-synuclein expression and pathology.

Response: To address this issue, we have performed co-culture analysis and provided the results in Figure S14. In this experiment, primary neurons were treated with wtNBD or mNBD and following that the neurons were kept with co-culture inserts containing PFF-induced primary microglia. At different time points following co-culturing, we measured α -syn mRNA and protein expression from neurons to show the effect of NBD peptides on the expression of α -syn.

Page 17, "Therefore, glial activation has an important role to play in α -syn spreading.": No direct evidence for this statement was provided.

Response: We completely agree with the reviewer on this comment as we did not show that glial activation directly impacts α -syn spreading. To address this issue, we used the approach of microglia-neuron coculture system, where following PFF-treatment in microglia we found increased expression of α -syn in neurons and that induction is significantly inhibited by wtTIDM-mediated inhibition of TLR2 in microglia. It asserts that TLR2 activation and resultant increase in inflammatory molecules by microglia have positive impact on neuronal α -syn expression. Spreading and amplification of α -syn aggregates happen when newly formed monomeric α -syn associates with existing fibrils and propagate the signal from neuron to neuron. In this process when α -syn species are released from one neuron, it binds to neighboring glial cells and induces inflammatory response. In the present investigation, we demonstrate that PFF-induced microglial activation and release of inflammatory cytokines further act on other neurons and up-regulate α -syn expression via the NF- κ B pathway. Therefore, glial activation plays major role in increasing α -syn expression, which further promotes spreading of existing α -syn aggregates.

Reviewer #3:

The current study leverages the wtTIDM (TLR2-interacting domain of MyD88 peptide), which was previously developed in the author's lab, to treatment various synucleinopathy mouse models and reports the attenuation of PFF-induced microglial inflammation, decrease in alpha-synuclein spreading, and neuroprotection. The study also shows that a few proinflammatory cytokines increase the transcription of alpha-synuclein gene in neurons through NF- κ B activation, which is inhibited by NEMO-binding domain peptide (wtNBD). While the notion of TLR2-MyD88-NF- κ B in the regulation of alpha-synuclein-associated neuroinflammation and spreading has previously been shown, the increase of alpha-synuclein expression induced by proinflammatory molecules in neurons is interesting and novel. However, there are multiple aspects of the reports that need additional investigations in order to support the main conclusions.

Response: Thanks for the enthusiasm.

The main concerns are the following:

1. For both peptides that are administrated in mice, the bioavailability and target engagement are poorly or not characterized, therefore, the action of the peptides is unclear and the results are not convincing. For example, what is the evidence for the wtTIDM penetration of BBB and disruption of TLR2-MyD88 interaction in the mouse brains?

Response: We are thankful to the reviewer for making such an important comment about the bioavailability of the peptides in brain. Recently we have demonstrated that after intranasal administration, wtTIDM peptide enters into the hippocampus of 5XFAD mice (Pahan lab, 2018, J. Clin. Invest., 128: 4297-4312). Please see Figure S10 of this paper for Alexa 680-labeled wtTIDM and Figure 5A-C for quantification by ESI-MS.

To address this here, we have added Figure S4 where we have shown that after intranasal administration, both wtTIDM and mTIDM peptides can reach the brain.

2. It is difficult to understand what the exact role TLR2 receptor plays for PFF-synuclein phagocytosis. On the one hand, TLR2 is activated with PFF treatment; on the other hand, disruption of TLR2-MyD88 interaction does not alter the PFF phagocytosis. Have the authors tried chemical inhibitors that block phagocytosis to validate the conclusion of unaltered phagocytosis?

Response: From our experiments, it is well evidenced that absence of TLR2 in TLR2 (-/-) mice hampers microglial phagocytosis and that is also supported by earlier report (Kim et al., 2013). In contrast blocking TLR2-MyD88 interaction by wtTIDM peptide does not inhibit phagocytosis, but prevents α -syn PFF-induced induction of inflammation in microglia. To prove that TIDM treatment does not interfere with basal or stimulated phagocytosis and to avoid confusion regarding phagocytosis of α -syn, we also conducted this test using latex beads and that also showed unaltered phagocytosis. It indicates that phagocytosis is not dependent on the binding of TLR2 with MyD88 and that a TLR2-dependent, but MyD88-independent, signaling pathway is responsible for microglial phagocytosis.

3. The authors used MN9D and SH-SY5Y cell lines only to test the effect of proinflammatory cytokines in increasing alpha-synuclein expression. However, this is not entirely convincing for the conclusion as they are not real neurons. The authors are recommended to use at least primary neuronal cultures to verify the conclusions. Furthermore, the authors failed to see any increase of alpha-synuclein *in vivo* (Fig. 2C, 2K, 3D). Therefore, it is a direct conflict with the claim of increased expression of alpha-synuclein by proinflammatory cytokines.

Response: We are thankful again to the reviewer for such a valuable comment. To address this issue, we have conducted co-culture experiments including primary microglia and primary dopaminergic neurons to show that PFF-induction in microglia leads to generation of inflammatory molecules to increase neuronal α -syn expression via NF- κ B.

Secondly, we have observed conspicuous increase in α -syn monomer in detergent insoluble fractions *in vivo* from different brain regions after PFF-seeding; however that was not reflected in soluble fractions. It signifies that the newly synthesized α -syn monomers are more recruited in the α -syn aggregates or existing fibrils in the PFF-seeded brain and therefore the up-regulation in α -syn monomer level is convincingly seen in detergent insoluble fractions. However, to address the issue of

transcriptional up-regulation of α -syn, we performed in situ ChIP analysis and found enhanced binding of NF- κ B subunits, p300 and RNA Polymerase II to α -syn gene promoter. Altogether, it firmly suggests that α -syn expression is enhanced in PFF-seeded brains.

Additional concerns/comments:

1. In Immunoprecipitation data (Fig. 1C), the authors should perform the IgG IP and Myd88 IP at the same time. Also, they need to provide immunoblotting for PFF detection in Input section.

Response: We have done IgG IP and Myd88 IP at the same time. Please see Figure S1. As suggested by you, we also tried to detect PFF by immunoblotting as input. However, after several attempts, we remained unable to detect PFF. It is possible because for this experiment, we incubated BV-2 microglial cells with PFF for 1 h and within 1 h, PFF is not internalized in BV-2 cells. For phagocytosis of α -syn, microglia were incubated with PFF for 2 h (Fig. 1I-M).

2. To test the effect of TLR2 on synuclein uptake, the authors used GFP-tagged monomeric synuclein, which is irrelevant to PFF. The authors may use GFP-tagged PFF instead. This problem applies to Fig. 1I, J, K, and L.

Response: To address this issue we have used FITC-tagged latex beads. We could not use GFP-tagged PFF because PFF is known to inhibit microglial phagocytosis (Park et al., 2008).

3. In Fig 2B, the authors cut the blot between 15 and 30 kda to show monomeric and high-molecular weight of alpha-synuclein, separately. This is not convincing. It is highly recommended that the authors provide the whole gel as shown in Fig 2J.

Response: We are grateful to the reviewer for that suggestion. We have provided the whole blot this time for Figure 2B.

4. There is a lack of detailed information regarding the imaging quantification. For example, it is unclear how the authors quantified p-S129 synuclein signal in Fig 2F,2H,3G,3I. Did the authors quantify the signals per cell? What does each dot in the graph indicate? How did the authors exclude the decrease of pS129-synuclein signal from the death of dopaminergic neurons, which occurs in PFF-model? Besides, images of A53T+PFF+mTIDM condition shows intense p-S129 synuclein signal (Fig 2E, G), but does not match the quantification graph.

Response: We apologize for not elaborating the quantification method of pSyn129 intensity in details. We have quantified the intensity (stained with DAB) using Fiji (ImageJ2). All figures were deconvoluted to achieve H-DAB stained images and then each cell was outlined and the intensity was measured by the Analyze-measure option provided in the software. The mean of stained cells in A53T brains was compared to that of cells of nTg brain. Darker staining means lower is the mean value. The mean value of white is considered to be highest (255), and therefore the formula used for calculating relative O.D. is $\log_{10}(255/\text{mean of each cell})$. In that way, we measured the values from two sections (15 cells/section) of each brain and the average value obtained from each section is shown in the figures 2F, 2H, 3G, 3I. Therefore, if n=5 for the immunostaining, we obtained 10 different values representing 10 different sections from 5 animals of a single group. We have also changed the representative image in Figure 2E to match the quantification.

5. The result showed more pS129-synuclein signals (Fig. 2E) in A53T+PFF group where dopaminergic neurons are dying (Fig. 4A). This is inconsistent with the previously published

data (Luk et al., 2016, Science), which showed a reduction of pS129-synuclein-positive cells due to the death of dopaminergic neurons. The authors should clarify this.

Response: It might not be the actual phenomenon happening in SN following PFF-seeding in striatum. In the report of Luk et al., clear presence of pS129-synuclein was shown in nigral dopaminergic neurons and there was significant neuronal death at 180 days post PFF-injection. This phenomenon is also recapitulated by several other studies (Mao et al., 2016; Yun et al., 2018) and therefore PFF-induced model is used as a sporadic model for Parkinson's disease.

6. It is recommended to use “ $\mu\text{g/ml}$ ” unit, not “ μM ”, to describe the concentration of PFF.

Response: As per the suggestion, we have also mentioned the concentration of PFF in $\mu\text{g/ml}$ format in the figure legends of Figure 1 and Figure 6.

7. In Fig 4, the claim for the loss of DA neurons based on counting of TH+ cells could be misleading. It is recommended to include Nissl staining to assure the loss of DA neurons rather than reduction of TH expression.

Response: We aimed to show the death of nigral dopaminergic neurons by staining with the dopaminergic neuronal marker TH, and this procedure is being practiced for the last several decades. Nissl staining would obviously stain all kinds of neurons in SN, however that data might not make it specific to Parkinson's disease. Therefore, we kept the data of TH neuronal counting to demonstrate dopaminergic neuronal loss.

References:

- Brahmachari, S., et al., 2016. Activation of tyrosine kinase c-Abl contributes to alpha-synuclein-induced neurodegeneration. *J Clin Invest.* 126, 2970-88.
- Kim, C., et al., 2013. Neuron-released oligomeric alpha-synuclein is an endogenous agonist of TLR2 for paracrine activation of microglia. *Nat Commun.* 4, 1562.
- Lau, A., et al., 2020. alpha-Synuclein strains target distinct brain regions and cell types. *Nat Neurosci.* 23, 21-31.
- Li, Y., et al., 2019. CXCL12 is involved in alpha-synuclein-triggered neuroinflammation of Parkinson's disease. *J Neuroinflammation.* 16, 263.
- Liu, G., et al., 2014. Aldehyde dehydrogenase 1 defines and protects a nigrostriatal dopaminergic neuron subpopulation. *J Clin Invest.* 124, 3032-46.
- Luk, K.C., et al., 2012. Pathological alpha-synuclein transmission initiates Parkinson-like neurodegeneration in nontransgenic mice. *Science.* 338, 949-53.
- Mao, X., et al., 2016. Pathological alpha-synuclein transmission initiated by binding lymphocyte-activation gene 3. *Science.* 353.
- Park, J.Y., et al., 2008. Microglial phagocytosis is enhanced by monomeric alpha-synuclein, not aggregated alpha-synuclein: implications for Parkinson's disease. *Glia.* 56, 1215-23.
- Rey, N.L., et al., 2019. alpha-Synuclein conformational strains spread, seed and target neuronal cells differentially after injection into the olfactory bulb. *Acta Neuropathol Commun.* 7, 221.
- Sorrentino, Z.A., et al., 2017. Intra-striatal injection of alpha-synuclein can lead to widespread synucleinopathy independent of neuroanatomic connectivity. *Mol Neurodegener.* 12, 40.
- Wu, J., et al., 2019. Distinct Connectivity and Functionality of Aldehyde Dehydrogenase 1a1-Positive Nigrostriatal Dopaminergic Neurons in Motor Learning. *Cell Rep.* 28, 1167-1181 e7.
- Yun, S.P., et al., 2018. Block of A1 astrocyte conversion by microglia is neuroprotective in models of Parkinson's disease. *Nat Med.* 24, 931-938.

We believe that constructive criticisms from highly qualified reviewers have strengthened our manuscript.

Reviewers' Comments:

Reviewer #1:

Remarks to the Author:

The revised manuscript is quite good and the NF κ B regulation of alpha-syn is very interesting and impactful. The addition of the co-culture experiments is particularly strong. The authors still claim that Iba-1 is a sufficient marker of microglia and TLR2 in vivo is microglia in origin. It is okay if the authors chose not to take the reviewer's advice and use more microglia-specific markers which should, indeed, work in this model despite the authors argument against this. It is also okay if the authors did not use a more microglia-specific manipulation in vivo. However, it should be relatively simple to revise the manuscript for accuracy to 1) reflect that Iba-1 is not a microglia-specific marker and will label other peripherally-derived immune cells and 2) other cells could be a source of TLR2 signaling in vivo such as astrocytes (e.g. PMID: 18768838) which also seem to be more reactive in their model according to Figure S6.

Reviewer #2:

Remarks to the Author:

The manuscript has been improved through the revision. However, there still are a few points that need clarification.

1. "Page 17, "Therefore, glial activation has an important role to play in α -syn spreading.": No direct evidence for this statement was provided.

Response: We completely agree with the reviewer on this comment as we did not show that glial activation directly impacts α -syn spreading. To address this issue, we used the approach of microglia-neuron coculture system, where following PFF-treatment in microglia we found increased expression of α -syn in neurons and that induction is significantly inhibited by wtTIDM-mediated inhibition of TLR2 in microglia. It asserts that TLR2 activation and resultant increase in inflammatory molecules by microglia have positive impact on neuronal α -syn expression. Spreading and amplification of α -syn aggregates happen when newly formed monomeric α -syn associates with existing fibrils and propagate the signal from neuron to neuron. In this process when α -syn species are released from one neuron, it binds to neighboring glial cells and induces inflammatory response. In the present investigation, we demonstrate that PFF-induced microglial activation and release of inflammatory cytokines further act on other neurons and up-regulate α -syn expression via the NF- κ B pathway. Therefore, glial activation plays major role in increasing α -syn expression, which further promotes spreading of existing α -syn aggregates."

Comment: I understand that glial activation can induce the expression of a-synuclein. However, this is a completely different issue from a-synuclein spreading. Unless the authors provide the direct evidence for a-synuclein spreading, this claim must be deleted from the manuscript.

2. Fig. 4: Judging by the extent of TH cell loss in the SN and Th reduction in the striatum, it is unlikely that the observed behavioral phenotypes are rooted from dopaminergic degeneration. This point needs to be discussed/clarified.

3. Fig. S14 E: the results should be quantified.

Reviewer #3:

Remarks to the Author:

The authors have addressed most of my concerns, except the TH+ staining for the indication of dopamine neuron viability. It is true that many publications simply performed counting of TH+ cells and drew conclusions of degeneration based on reduction of the number of TH+ cells. However, such an assay cannot exclude the possibility of reduced expression of TH in neurons (as

opposed to loss of neurons) therefore the those conclusions are overstated and inaccurate. If the authors cannot provide an independent measurement for validating loss of dopamine neurons, the authors should modify the conclusions.

Reviewer #1:

Comment: *“The revised manuscript is quite good and the NFkappaB regulation of alpha-syn is very interesting and impactful. The addition of the co-culture experiments is particularly strong. The authors still claim that Iba-1 is a sufficient marker of microglia and TLR2 in vivo is microglia in origin. It is okay if the authors chose not to take the reviewer's advice and use more microglia-specific markers which should, indeed, work in this model despite the authors argument against this. It is also okay if the authors did not use a more microglia-specific manipulation in vivo. However, it should be relatively simple to revise the manuscript for accuracy to 1) reflect that Iba-1 is not a microglia-specific marker and will label other peripherally-derived immune cells and 2) other cells could be a source of TLR2 signaling in vivo such as astrocytes (e.g. PMID: 18768838) which also seem to be more reactive in their model according to Figure S6.”*

Response: We thank the reviewer for recognizing our effort to modify the manuscript and the associated data. As per reviewer’s valuable suggestions, we have performed immunofluorescence analysis using microglia specific marker P2RY12 in two different experiments: Firstly, we co-immunostained with TLR2 in non-Tg and 9-month old A53T brain sections to evaluate up-regulation of TLR2 specifically in brain microglia (Figure S3E-S3F). Secondly, we conducted double immunostaining of P2RY12 and iNOS in A53T PFF-seeded A53T animals in order to examine the induction of microgliosis in these brains and also to confirm the effect of wtTIDM on inhibition of microglial activation (Figure S6A-S6B).

In addition, we have also confirmed astroglial TLR2 up-regulation in aged A53T brains and discussed the possible outcome of astroglial TLR2 activation. Supporting the earlier notion, it can be stated that astrocytic activation in PFF-induced mice might happen directly by PFF (where TLR2 is possibly involved in conducting PFF-induced astroglial inflammation) and by microglia derived inflammatory molecules, which were previously shown by many studies. As the effect of A1 astrocytic activation on pathological α -syn accumulation and dopaminergic neuronal death has been shown by Yun and co-workers, it is also possible that the expression of neuronal α -syn will increase by astrocyte-derived factors if these molecules induce NF- κ B activation in neurons. However, that hypothesis needs further investigation and is not covered in the present study.

Reviewer #2:

The manuscript has been improved through the revision. However, there still are a few points that need clarification.

Comment 1. *“Page 17, “Therefore, glial activation has an important role to play in α -syn spreading.”: No direct evidence for this statement was provided.*

Response: We completely agree with the reviewer on this comment as we did not show that glial activation directly impacts α -syn spreading. To address this issue, we used the approach of microglia-neuron coculture system, where following PFF-treatment in microglia we found increased expression of α -syn in neurons and that induction is significantly inhibited by wtTIDM-mediated inhibition of TLR2 in microglia. It asserts that TLR2 activation and resultant increase in inflammatory molecules by microglia have positive impact on neuronal α -syn expression. Spreading and amplification of α -syn aggregates happen when newly formed

monomeric α -syn associates with existing fibrils and propagate the signal from neuron to neuron. In this process when α -syn species are released from one neuron, it binds to neighboring glial cells and induces inflammatory response. In the present investigation, we demonstrate that PFF-induced microglial activation and release of inflammatory cytokines further act on other neurons and up-regulate α -syn expression via the NF- κ B pathway. Therefore, glial activation plays major role in increasing α -syn expression, which further promotes spreading of existing α -syn aggregates.”

Comment: I understand that glial activation can induce the expression of α -synuclein. However, this is a completely different issue from α -synuclein spreading. Unless the authors provide the direct evidence for α -synuclein spreading, this claim must be deleted from the manuscript.

Response: Thanks for the enthusiasm. We completely agree with the reviewer that our findings clearly demonstrate inflammatory molecules induced up-regulation of α -syn expression, whereas the direct evidence of α -syn spreading is currently not provided in the manuscript. Therefore, as per the valuable advice from the reviewer, we have modified our conclusions in the discussion stating that glial activation induces transcriptional up-regulation of α -syn via NF- κ B pathway. Perhaps this might be the first report demonstrating NF- κ B-dependent α -syn expression by inflammatory mediators.

Comment 2. *“Fig. 4: Judging by the extent of TH cell loss in the SN and Th reduction in the striatum, it is unlikely that the observed behavioral phenotypes are rooted from dopaminergic degeneration. This point needs to be discussed/clarified.”*

Response: We completely agree with the critical comment made by the reviewer that the observed behavioral phenotypes might not be solely rooted from dopaminergic degeneration. Perhaps significant reduction in normal horizontal motor abilities are reproduced in more acute models of Parkinson's disease, such as acute MPTP-induced mice where almost 80% dopamine loss are found in striatum. In contrast, in the PFF-induced A53T mice we observed 40-50% loss of TH fibers and dopamine in striatum. This extent of dopamine loss might not cause such massive reduction of normal movements of animals, but it has been shown to correlate well with deficits in fore limb and hind limb usage of animals as revealed in rotarod test, rearing, pole tests and also in grip tests (Mao et al., 2016; Yun et al., 2018), which require fine tuning in motor coordination. In this study, for the PFF-induced animals, we have shown the data from rotarod test and rearing, confirming the deficits in motor skills in these animals. However, the data concerning the distance and velocity of animals in the open field arena are completely based on the live observations of animal movements. We observed that majority of the PFF-seeded animals are spending more time around the corner of the arena and exploring less in the center, resulting in overall less distance covered or cumulative duration of moving and it might not be a result of dopamine depletion. Therefore, considering the existing reports and as per reviewer's valuable suggestion, we are removing the open field behavioral parameters from Figure 4 and from Figure S18.

Comment 3. *“Fig. S14 E: the results should be quantified.”*

Response: We have added the quantification of these data in Figure S16F.

Reviewer #3:

Comment: *“The authors have addressed most of my concerns, except the TH+ staining for the indication of dopamine neuron viability. It is true that many publications simply performed counting of TH+ cells and drew conclusions of degeneration based on reduction of the number of TH+ cells. However, such an assay cannot exclude the possibility of reduced expression of TH in neurons (as opposed to loss of neurons) therefore the those conclusions are overstated and inaccurate. If the authors cannot provide an independent measurement for validating loss of dopamine neurons, the authors should modify the conclusions.”*

Response: We are thankful to the reviewer for the valuable suggestions, which helped us to strengthen our findings. We also agree on the point that following other studies on this line of research, we have addressed nigral dopaminergic neuronal death by counting the existing TH positive neurons, however the possibility of reduced TH expression still remains. Therefore, to clarify the reduction of TH expression, we have measured the mean fluorescence intensity of TH in viable neurons of SN, which showed significant down-regulation of TH level in dopaminergic neurons of PFF-induced brains (Figure S9B & S9E). Moreover, to find out the possibility of induction of neuronal death by PFF, we have performed cleaved caspase 3 immunostaining in nigral sections, co-stained with TH (Figure S9B) and with LB509 antibody recognizing aggregated α -syn (Figure S9A). Our findings confirmed that cleaved caspase 3 expression is triggered in PFF-induced brains and it was significantly less in animals receiving wtTIDM administration. At least, this finding indicates that the surviving TH neurons and LB509 positive cells are undergoing cell death cascade, whereas intranasal wtTIDM delays the induction of neuronal demise. Perhaps greater loss of TH neurons would have been obtained if the end point of the study was made 4 or 5 months following PFF-injection.

Reviewers' Comments:

Reviewer #3:

Remarks to the Author:

The authors have addressed my concerns. No more question.